# Selection of calibration events for modelling green urban drainage

Ico Broekhuizen[1], Günther Leonhardt[1], Jiri Marsalek[1], and Maria Viklander[1]

[1]Luleå University of Technology, Department of Civil, Environmental and Natural Resources Engineering, Urban Water Engineering. Luleå, Sweden

**Correspondence:** Ico Broekhuizen (ico.broekhuizen@ltu.se)

**Abstract.** Calibration of urban drainage models is typically performed based on a limited number of observed rainfall-runoff events, which may be selected from a larger dataset in different ways. In this study, 14 single- and two-stage strategies for selecting the calibration events were tested in calibration of a SWMM model of a predominantly green urban area. The event selection was considered in relation to runoff contributions from green pervious areas and such sources of uncertainty such as rainfall/runoff measurement uncertainties and catchment discretization. Even though all 14 strategies resulted in successful model calibration, the difference between the best and worst strategies reached 0.2 in Nash-Sutcliffe Efficiency (NSE) and the calibrated parameter values notably varied. Most, but not all, calibration strategies were robust to perturbations in calibration data and the use of a coarse catchment discretization model in the calibration phase. The various calibration strategies satisfactorily predicted 7 to 13 out of 19 validation events. The two-stage strategies performed better than the single-stage strategies when: (1) perturbing flow data in the calibration events by +-40%; and (2) using a coarser catchment discretization, especially in terms of total flow volume and peak flow rates. The two calibration strategies that performed the best in the validation phase were two-stage strategies. The findings in this paper show that various strategies for selecting calibration events lead in some cases to different results in the validation phase, and that calibrating impervious and green area parameters in two separate steps may increase the effectiveness of model calibration/validation by reducing the computational demand in the calibration phase and improving model performance in the validation phase.

*Copyright statement.* TEXT

## 1 Introduction

Calibration of generic urban drainage model codes is usually required to obtain a model representing an actual site with sufficient accuracy. In the calibration process, the information contained in records of relevant variables, such as rainfall and flow rates at the catchment outlet, is used for estimating model parameter values that produce results consistent with the data (Mancipe-Munoz et al., 2014). It can be expected that the best parameter estimates will be obtained when they are inferred from the largest amount of information, i.e. by using all data from a long series of measurements. However, the availability of calibration data may be limited and the nature of the calibration process, by trial and error, requires model iterations for many different parameter sets, which means that the runtime of the model has to be kept short and the length of the simulated periods

should be limited. Therefore, calibration may have to be performed on a limited number of rainfall events from a longer record. As each of the available events will differ from the others, it can be expected that the choice of a specific event (or an event set) will influence the results of calibration (Tscheikner-Gratl et al., 2016).

Tscheikner-Gratl et al. (2016) studied such influence by calibrating water level in the outflow pipe of a catchment using ten different rain events. They found that two of them could not be reproduced in calibration and the others, while successful in calibration, could only predict up to six of the remaining events. When applying the calibrated models with design storms, they found that the calibrated models predicted different flooding volumes. In calibration of combined sewer overflow (CSO) volumes, Kleidorfer et al. (2009b) compared calibration results obtained for (1) the five longest duration events and (2) the five highest peak flow events, finding that using the longest duration events reduced the number of measurement sites required for successful calibration. Schütze et al. (2002) demonstrated that calibration based on discrete events saved time compared to calibrating for a complete time series, but also that this introduced additional uncertainty. Mourad et al. (2005) showed that calibration of a stormwater quality model was sensitive to: (1) which randomly selected events were used, and (2) how many events were used.

While the above papers helped elucidate some aspects of the sensitivity of urban drainage model calibration to the calibration events used, such findings possess some limitations: firstly, only a limited number of generally available options for selecting calibration events has been considered; secondly, the modelling focused on traditional urban drainage systems, in which generation of runoff is dominated by impervious surfaces, but the current trend towards green urban drainage infrastructure creates the need to pay more attention to runoff processes on green areas (Fletcher et al., 2013). This second aspect also applies to investigations into other sources of uncertainty in urban drainage modelling, some of which have been investigated before, e.g. input and calibration data uncertainties(Dotto et al., 2014; Kleidorfer et al., 2009a) and spatial model resolution (Krebs et al., 2014; Petrucci and Bonhomme, 2014; Sun et al., 2014). However, these investigations used predominantly impervious catchments and it is, therefore, unknown to what extent their findings apply to greener urban catchments as well and how sensitive such results are to the calibration data set that was used.

Considering the above findings, the primary objective of the paper that follows is to advance the knowledge of calibration processes for green urban areas by examining different strategies for selecting calibration events and assessing the effects of such selections on the performance of a calibrated hydrodynamic model of a predominantly green urban catchment. Part of this is a proposal for a practical two-stage calibration strategy. Two secondary objectives are to verify: (1) the findings from previous urban drainage modelling studies on a greener (less impervious) catchment, and (2) sensitivity of the earlier findings to the calibration data used.

## 2 Materials and methods

### 2.1 Study site and data

The study site is a 10.2 ha catchment in the city of Luleå, Sweden (see Figure 1). The catchment area comprises 63% of green areas, 12% of impervious areas connected directly to the storm sewer system, and 25% of impervious areas draining onto

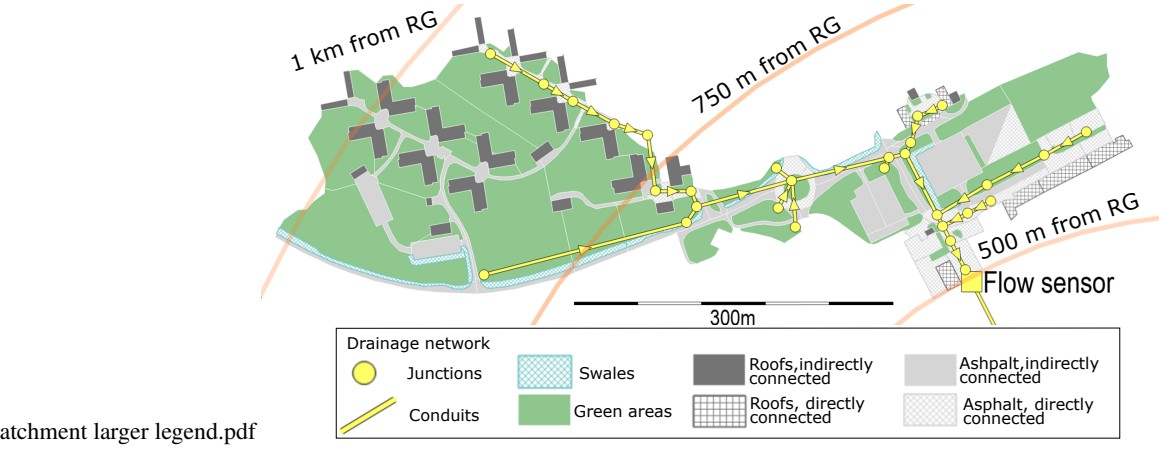

catchment larger legend.pdf

**Figure 1.** Map of the studied catchment showing elements of the high-resolution rainfall-runoff model and the distance of the catchment to the rain gauge (RG). The diameters of the pipes range from 400 mm for the main trunk where the flow sensor is located to 200 mm for the smaller branches.

adjacent green areas. The green areas include a number of vegetated swales that are connected to the storm sewer system at their lowest point.

Precipitation was measured at 1-minute intervals with a Geonor T200B weighing-bucket precipitation gauge located outside of the study catchment, about 500 and 1,000 metres from the nearest and furthest borders of the catchment, respectively (see circles in Figure 1). The gauge was tested in the field and confirmed to work well twice a year in 2016 and 2017, and before 2016, such tests were also performed occasionally. Laboratory and field tests (by others) found this design of precipitation sensor to be a reliable instrument (Duchon, 2002; Lanza et al., 2010). Records were available for individual rain events in 2013-2015 and continuously for 2016 and 2017.

Flow rates in the storm sewer draining the catchment were measured at 1-minute intervals by means of an ISCO 2150 AV sensor (a combination of an acoustic Doppler velocimeter and a pressure transducer) installed in the catchment outlet formed by a 400 mm diameter concrete sewer pipe. This type of sensor was assessed in the laboratory by Aguilar et al. (2016) and found to have a combined uncertainty (consisting of bias, precision and benchmark uncertainty) of ±19.0 mm for the water depth measurements (the test range was 10-150 mm) and ±0.0985 m/s for the velocity measurement (test range 0.1-0.6 m/s). These tests were carried out in a 0.46 m wide square channel, so the stage-discharge relationship was different from the study site described herein. It was also reported that the field performance of this type of sensors can suffer from the presence of too few (Teledyne ISCO, 2010) or too many particles suspended in the water (Nord et al., 2014).

While the difficulties in estimating all the uncertainties at the actual field site prevented a precise determination of the uncertainties' magnitude, the general lab tests of the sensors used confirmed the acceptability of their records for the study purpose. Finally, it was also confirmed by Dotto et al. (2014) that errors in the calibration data can be compensated for in the calibration process.

The available precipitation record was divided into rainfall events with a minimum inter-event time of no precipitation of six hours. Events deemed suitable for use in calibration were selected using the following criteria:

1. A minimum total precipitation of 2 mm (Hernebring, 2006).

2. No or small gaps in rain and flow data , i.e. both have to be available for >90% of the event duration.

3. Sufficient in-pipe water depths for the flow sensor to work reliably: >10 mm during at least 50% of the event and >25 mm at least once in the event, based on recommendations from the manufacturer (Teledyne ISCO, 2010).

4. Peak flow $>2$ L s$^{-1}$, since relative measurement uncertainties are high below this point.

5. No snowfall or -melt, since these would introduce additional processes in the hydrological behaviour and model of the catchment.

Calibration and validation periods were separated by using the 19 observed events from 2016 for the validation period, and the 32 events from 2013-2015 and 2017 for the calibration period. In this way, all the calibration scenarios (see section 2.3) were tested (validated) against the same dataset and no calibration scenarios could benefit from including calibration events that also appeared in the validation set. The year 2016 was selected as the validation period for two reasons: it was the year with total precipitation closest to the annual mean, and the measured data records were continuous. Table 1 contains an overview of all events that were used in at least one calibration scenario as well as an initial estimate of the runoff from green areas.

## 2.2 Runoff model and calibration approach

The US EPA Storm Water Management Model (SWMM) was selected since it is a commonly used semi-distributed urban drainage model that allows to route runoff from one sub-catchment to another. This routing feature was needed since it allows for a high-resolution model setup in which each subcatchment (146 were used in total) features a single land cover. The high resolution input data needed for this approach was available in the form of GIS data, aerial photographs, and observations from site visits. The advantage of these single land-cover subcatchments is that their parameter values maintain their physical meaning and can be calibrated (or appropriate values found in the literature) for each land use or cover. The traditional approach of using larger subcatchments with multiple land uses/covers usually necessitates calibration to estimate the values of parameters that then represent a weighted average value over multiple land uses/covers. Some spatial characteristics, such as the slope and the width of subcatchments, can also be estimated more easily for smaller, uniform subcatchments. This approach has been used successfully by e.g. Krebs et al. (2014, 2016), Petrucci and Bonhomme (2014) and Sun et al. (2014). Within SWMM the Green-Ampt infiltration method was selected since it can be calibrated with just two parameters (Rossman, 2016).

Whenever feasible, parameters for different subcatchments were set directly from the available GIS data and site visits, i.e. the sizes and slopes of all subcatchments and sewer pipes, as well as the catchment widths of small and disconnected roofs. For other subcatchments the catchment width was calibrated together with the other model parameters. To reduce the scope of the calibration problem, parameters were grouped based on land cover, yielding a total of thirteen calibration parameters for the

**Table 1.** Characteristics of all rainfall events used in one or more calibration scenarios.

| Event # | Precipitation sum in preceding 72 hr | Precipitation sum (P_sum) | Precipitation duration (D_prec) | Average precipitation intensity (PI_mean) | Highest 30-minute average precipitation intensity (PI_30m) | Runoff volume (QV) | Percentage runoff (QV_ppP) | Peak flow rate (Q_max) | Highest 60-minute average flow rate (Q_60m) | Runoff from green areas [a] | Of which originating from imperv. areas [b] | Originating from green areas [c] | Average percentage runoff from green areas [d] |
|---|---|---|---|---|---|---|---|---|---|---|---|---|---|
| | mm | mm | hr | mm hr$^{-1}$ | mm hr$^{-1}$ | mm | % | L s$^{-1}$ | L s$^{-1}$ | mm | mm | mm | % |
| 199 | 2.4 | 13.8 | 41.6 | 0.3 | 4.0 | 1.7 | 12.4 | 4.2 | 3.3 | 0.06 | 0.02 | 0.04 | 0.3 |
| 209 | 0.2 | 8.0 | 9.5 | 0.8 | 2.8 | 0.5 | 6.9 | 4.5 | 2.7 | | | | |
| 211 | 8.3 | 9.7 | 22.8 | 0.4 | 6.9 | 1.1 | 11.1 | 29.2 | 11.1 | | | | |
| 214 | 7.3 | 6.4 | 12.1 | 0.5 | 4.3 | 0.6 | 10.1 | 40.5 | 8.5 | | | | |
| 222 | 1.1 | 9.8 | 12.8 | 0.8 | 7.5 | 0.7 | 7.2 | 26.4 | 13.3 | | | | |
| 270 | 0.0 | 9.3 | 38.5 | 0.2 | 3.5 | 1.1 | 11.3 | 22.9 | 8.7 | | | | |
| 306 | 10.1 | 8.6 | 9.1 | 0.9 | 7.1 | 0.7 | 8.5 | 27.5 | 9.3 | | | | |
| 307 | 18.3 | 29.9 | 37.7 | 0.8 | 8.5 | 4.9 | 16.2 | 71.2 | 42.9 | 1.27 | 0.36 | 0.91 | 3.0 |
| 310 | 12.7 | 8.6 | 10.0 | 0.9 | 7.5 | 1.2 | 14.0 | 37.4 | 17.4 | 0.17 | 0.05 | 0.12 | 1.4 |
| 530 | 13.8 | 6.7 | 2.8 | 2.4 | 7.2 | 0.8 | 11.2 | 58.9 | 13.5 | | | | |
| 939 | 0.6 | 7.0 | 25.6 | 0.3 | 1.0 | 0.4 | 5.7 | 2.1 | 1.8 | | | | |
| 962 | 0.0 | 8.5 | 11.2 | 0.8 | 1.4 | 2.1 | 24.9 | 4.9 | 4.4 | 1.09 | 0.31 | 0.78 | 9.2 |
| 971 | 0.2 | 2.6 | 18.6 | 0.1 | 1.1 | 0.3 | 11.3 | 4.0 | 2.9 | | | | |
| 978 | 12.7 | 25.0 | 65.8 | 0.4 | 5.8 | 4.8 | 19.1 | 64.5 | 16.6 | 1.77 | 0.50 | 1.27 | 5.1 |
| 982 | 0.0 | 5.6 | 3.4 | 1.7 | 7.0 | 0.9 | 15.8 | 49.5 | 17.2 | 0.21 | 0.06 | 0.15 | 2.7 |
| 984 | 13.1 | 2.4 | 6.3 | 0.4 | 4.6 | 1.4 | 59.1 | 71.7 | 14.0 | 1.12 | 0.32 | 0.80 | 33.7 |
| 995 | 4.8 | 2.1 | 8.5 | 0.2 | 1.8 | 0.6 | 28.6 | 32.0 | 9.7 | 0.35 | 0.10 | 0.25 | 11.9 |
| 997 | 2.2 | 24.6 | 49.0 | 0.5 | 2.4 | 5.1 | 20.7 | 15.0 | 6.9 | 2.14 | 0.61 | 1.53 | 6.2 |
| 1001 | 0.0 | 35.3 | 56.6 | 0.6 | 8.6 | 8.8 | 25.0 | 56.5 | 32.5 | 4.58 | 1.30 | 3.28 | 9.3 |
| 1004 | 22.5 | 4.2 | 13.9 | 0.3 | 5.9 | 1.1 | 25.2 | 33.3 | 10.6 | 0.56 | 0.16 | 0.40 | 9.5 |
| 1019 | 0.5 | 22.3 | 49.7 | 0.4 | 2.3 | 4.7 | 21.2 | 12.9 | 9.3 | 2.06 | 0.58 | 1.47 | 6.6 |
| 1028 | 6.2 | 2.8 | 7.0 | 0.4 | 1.3 | 1.2 | 43.5 | 6.3 | 4.2 | 0.89 | 0.25 | 0.64 | 22.5 |

[a] Calculated assuming 100% runoff from impervious areas: a = QV – 0.12 P_sum, where 0.12 is the percentage of directly connected impervious area. (Some of this runoff originated from impervious areas that drained to green areas).

[b] Calculated as b = a (25 / (25+63)), where 25 and 63 are the percentages of indirectly connected impervious surfaces and green surfaces respectively.

[c] Calculated as c = a – b

[d] Calculated as d = c / P_sum

**Table 2.** Calibration parameters and their ranges.

| Parameter | Abbr. | Groups | Range | Reference |
|---|---|---|---|---|
| | | Asphalt parking lots (AP) | 20-200 | |
| Subcatchment width [m] | width | Grass areas (GR) | 1-200 | |
| | | Swales (SW) | 0-5 | Physical dimensions of subcatchments |
| Subcatchment length [m] | length | Asphalt roads[a] | 0.5-5 | |
| | | Impervious surfaces (IMP) | 0.005 – 0.015 | |
| | | Grass areas (GR) | 0.1 – 0.5 | |
| Manning's number [-] | n | Swales (SW) | 0.1 – 0.5 | |
| | | Pipes | 0.010 – 0.015 | (Krebs et al., 2016; Rossman, 2016) |
| | | Impervious surfaces (IMP) | 0 – 2.5 | |
| Depression storage [mm] | s | Grass areas (GR)[b] | 0 – 20 | |
| | | Swales (SW)[c] | 0 – 150 | (Rujner et al., 2018)[d] |
| Saturated hydraulic conductivity [mm hr$^{-1}$] | ksat | Grass areas (GR)[e] | 1 - 200 | (Rawls et al., 1983) |
| Initial moisture deficit [-] | imd | Grass areas (GR)[e] | 0.10 – 0.35 | |

[a] In SWMM, the subcatchment width is an input, but in this group of subcatchments, the length (in the flow direction) showed more similarity among the subcatchments, so it was calibrated instead of the width.

[b] Includes vegetation and trees as well.

[c] The maximum value was intentionally set high since the swales' outlets are not always located exactly at the lowest points and the swales can be observed with larger ponds after heavy rain events.

[d] Field experiments on similar swales in the same city.

[e] Used for both grass areas and swales.

hydrodynamic model. Parameter values were limited based on values reported in the literature (see Table 2). The precipitation gauge was situated a few hundred metres outside of the actual catchment, and may have provided a biased estimate of the catchment rainfall. Therefore, a rainfall multiplier for each individual rainfall event was included in the calibration. This approach has been used with satisfactory results e.g. by Datta and Bolisetti (2016), Fuentes-Andino et al. (2017, and Vrugt

5 et al. (2008), although it is limited by assuming a simple multiplicative difference between the gauge and catchment-average rainfall, which is not necessarily the case (Del Giudice et al., 2016). Furthermore, rainfall multipliers do not address the spatial variability of the rainfall, but in the absence of multiple rain gauges or other information about the spatial variability of rainfall in the study catchment, there were no feasible alternatives in this case. The rainfall multipliers create a way of adjusting the rainfall volume in the calibration so that the simulated runoff volume can better match the observed runoff volume. However,

10 the multipliers do not allow distinguishing between (1) deviations between rainfall at the gauge and the catchment-averaged rainfall, (2) errors in the rainfall measurement, and (3) errors in the runoff measurement. A more traditional approach would be to calibrate the percentage of impervious areas, but in view of the availability of high-resolution land-cover information, it was preferred to apply rainfall multipliers instead.

Green surfaces like those in the study area have a long hydrological memory for antecedent rainfall, and this had to be accounted for in the simulations. Neglecting this memory would increase the risk of green areas allowing unrealistically high infiltration in some rainfall events. Since SWMM does not allow for setting the initial values of state variables directly, such adjustments can be done by choosing an appropriate warm-up period for modelling runs. When sufficiently long warm-up periods are used, this approach offers an advantage consisting of treating the first rainfall/runoff peak of an event the same as way as any following peaks, i.e., with initial conditions corresponding to a continuous simulation. The required length of this warm-up period was estimated by finding the last time before each rainfall event when the study area was dry. This was calculated for all rainfall events using the actual precipitation data and for various values for the maximum depression storage and infiltration rate. The last antecedent time when the study area was dry was then used as the starting point of the warm-up period. This lookup procedure was applied to every event for each iteration in the calibration process, so that all events were treated the same way as in a continuous simulation.

In the calibration process, the Shuffled Complex Evolution - University of Arizona algorithm (SCE-UA; Duan et al. (1994)) was used to estimate the optimal values of the parameters. The algorithm was selected because it is commonly used in hydrological studies and allows for parallel computing. The Python library SPOTPY (Houska et al., 2015), which includes this algorithm, was used to carry out the entire calibration process.

## 2.3 Event selection

This paper investigates single- and two-stage calibration scenarios (CS), with each CS using six rainfall events. The single-stage CSs used the six events with the highest values of a certain event characteristic, and calibrated all parameters simultaneously. Two-stage calibration scenarios calibrated first the parameters related to impervious areas, using a set of three rainfall events, followed by the pervious area parameters using another set of three rainfall events. Since only 12% of the total catchment surface is impervious and connected directly to storm sewers, it was assumed that the events, for which runoff volume was less than 12% of rainfall volume, produced runoff only from impervious areas. (It is conceivable that there is some contribution of green areas when the percentage runoff is less than 12%, and in that case the threshold should be set at a lower value, but since the amount of green area runoff and the appropriate value of the threshold would be highly dependent on antecedent conditions this was not included here.) Therefore, these events were suitable for calibration of impervious area parameters in the first stage of the calibration process. Following this step, events with more than 12% runoff were assumed to also include runoff from green areas and were used to estimate pervious area parameters in the second stage of the calibration. When calibrating the green area parameters, the parameters related to impervious areas were kept fixed at their values from the first stage. This procedure splits the optimization problem into two smaller problems that have fewer parameters and shorter run times. The smaller number of parameters (reduced dimensionality) can ease the search for optimal parameter sets, while the shorter run time per iteration allows shortening the total time needed, increasing the number of iterations used, or including more events in the calibration.

Characteristics related to the rainfall, flow depths and flow rates were calculated for each event. For the single-stage calibration scenarios, the six highest ranking events for each characteristic were selected. For the two-stage calibration scenarios,

the three highest ranking events with less than 12% runoff were selected for the first stage and the three highest ranking events with more than 12% runoff were selected for the second stage. Applying the calibrated rainfall multipliers in the calibration (Sect. 2.2) means that event properties relating to rainfall and percentage runoff will change, and the percentage runoff can change from <12% to >12% and vice versa. Doing this consistently for all events in the calibration procedure would require (1) re-calculating which events should be available in each stage, (2) estimating in some way rainfall multipliers for all events, including those not initially selected by any calibration scenario, (3) re-calculating which events are used in each CS, and (4) repeating the calibration for any CS that has had any of its events changed. Although this might improve the overall results of the proposed calibration procedure, it would also increase the complexity and raise several new issues, such as how to obtain a calibrated rainfall multiplier for the 10 events that were not used in any CS. We considered this to be beyond the paper's original scope of examining different strategies for calibration event selection and proposing a practically useable two-stage calibration procedure.

To avoid making the comparison too large in scope, a limited number of calibration scenarios (eight single-stage and six two-stage) was selected for use in this study. This selection was made so that it included a range of different characteristics and avoided multiple CSs with the exact same set-up of events. The names of the CSs consist of two or three elements:

- T6 (Top 6) for single-stage or T32S (Top 3 - 2 stages) for two-stage scenarios.

- The relevant event characteristic: precipitation (P), precipitation intensity (PI), runoff flow rate (Q), flow volume (QV), or flow volume as percentage of rain QV_ppP, precipitation duration D_prec.

- The duration over which the characteristics were calculated: sum, mean and max refer to the whole event. 30 and 60 min refer to the time interval used to calculate an average rainfall intensity or flow rate (i.e. the highest value found within the event for a 30 or 60 minute moving average). Calculating rainfall intensities and average flow rates over these windows rather than the entire event suppresses the effects of e.g. dry periods within events on such calculations.

The calibration scenario N_T6 consists of the six events that were selected most often in other calibration scenarios with the goal of obtaining a set of events that score highly on a variety of characteristics.

## 2.4 Other sources of uncertainty

Calibration data selection is not the only source of uncertainty in urban drainage modelling. Deletic et al. (2012) identified nine sources: (1) input data, (2) model parameters, (3) calibration data measurements, (4) calibration data selection, (5) calibration algorithm, (6) objective functions, (7) conceptualisation (e.g. discretization), (8) process equations and (9) numerical methods and boundaries. As described above, calibration data selection is the focus of this paper. However, earlier findings regarding the other sources of uncertainties were based on predominantly impervious catchments and they should not be assumed to apply equally to greener catchments. The nature of the catchment in this paper provides an opportunity to (1) check if these findings apply to greener catchments as well and (2) check if these findings are sensitive to the calibration data set that is used. It was beyond the scope of this paper to break new ground in all of the nine sources listed above; therefore, we focused on

uncertainty sources that have been covered in earlier literature. The uncertainties arising from objective functions, calibration algorithms and numerics are not considered explicitly in this paper. The choice of objective function can be expected to affect the calibration results, but this issue has received hardly any attention in urban drainage modelling, except for some short remarks by Barco et al. (Barco et al., 2008). Likewise, the calibration algorithm (Deletic et al., 2012; Houska et al., 2015) and
numerical issues (Deletic et al., 2012; Kavetski et al., 2006) are recognized as sources of uncertainty, but there is a lack of studies addressing these specifically for urban drainage modelling that could be referred to here. Since breaking new ground in these areas was considered beyond the scope of this paper, these sources of uncertainty are not considered here. The inclusion of other sources of uncertainty is described in the remainder of this section.

*Rainfall input uncertainty*. Earlier studies of the Geonor T200B rain gauge used have reported wind-induced undercatch of
4-5% (Duchon and Essenberg, 2001; Lanza et al., 2010). Additionally, there may be some deviations between the rainfall at the gauge and in the catchment. It is therefore possible that structural errors exist in the rainfall measurements. This aspect was investigated by examining the rainfall multipliers that were included for each event in the calibration (see Sect. 2.2). It should be noted that the rainfall multipliers are used to adjust flow volumes and that they may therefore also reflect uncertainties in e.g. subcatchment delineation and runoff routing.

*Parameter uncertainty*. The uncertainty of urban drainage model parameter estimates has been investigated extensively earlier, e.g., by Del Giudice et al. (2016), Dotto et al. (2009, 2011, 2012), Kleidorfer et al. (2009a) and Muleta et al. (2013). Therefore, this issue is addressed herein just by comparing the parameter values obtained in different calibration scenarios.

*Calibration data measurement uncertainties*. Measurement uncertainties of flow rates in storm sewer pipes have been described by a number of researchers, e.g., Aguilar et al. (2016), Blake and Packman (2008), Bonakdari and Zinatizadeh (2011),
Heiner and Vermeyen (2012), Lepot et al. (2014), Maheepala et al. (2001). In this paper, structural flow measurement errors are considered by testing calibration after reducing or increasing all flow observations by 40%. This value was chosen on the basis of uncertainties reported by Aguilar et al. (2016) and applied to the study outflow measurement location. . This is a rather simple approach and other ways of simulating errors in the measured data may be considered: e.g. Dotto et al. (2014) also tested the effect of random errors; However, since many different ways of perturbing flow data can be used it was deemed outside of
the scope of this paper to examine them all, and only the constant offset was used as a simple way of introducing errors in the flow measurement. However, it should be noted that the use of measured flow rates, implemented in this study, involves the presence of random errors in the calibration data sets used. The flow data from the validation period was not adjusted.

*Conceptualisation / model discretization*. Although model structure is also a recognized source of uncertainty (Deletic et al., 2012), it was not considered here since: (a) there is a lack of previous research on this topic for urban drainage modelling that
could be referred to, and (b) there is a lack of methods to address this issue, other than using different models in parallel, which was considered outside the scope of this study, and would be difficult since the catchment model requires some SWMM features (e.g. routing runoff from one subcatchment to another, good support for automated runs), which are not always available in other models.

The choice of catchment discretization into the subcatchments in the model has been investigated by several authors.
Tscheikner-Gratl et al. (2016) found that a lumped model was not able to reproduce the shapes of storm runoff hydrographs

**Table 3.** Calibration parameters and their ranges for the low-resolution model.

| Parameter | Abbr. | Groups | Range | Reference |
|---|---|---|---|---|
| Subcatchment width [m] | width | 5 individual subcatchments | 20 – 200 | Physical dimensions of subcatchments |
| Manning's coefficient [-] | n | Impervious surfaces (IMP) | 0.005 – 0.015 | |
| | | Pervious surfaces (GR) | 0.1 – 0.5 | |
| | | Pipes | 0.010 – 0.015 | (Krebs et al., 2016; Rossman, 2016) |
| Depression storage | s | Impervious surfaces (IMP) | 0 – 2.5 | |
| | | Pervious surfaces (GR) | 0 – 20 | |
| Percentage runoff routed from impervious to pervious (%) | | See footnote [a] | 1-99 | |
| Saturated hydraulic conductivity [mm hr$^{-1}$] | ksat | Grass areas (GR) | 1 - 200 | (Rawls et al., 1983) |
| Initial moisture deficit [-] | imd | Grass areas (GR) | 0.10 – 0.35 | |

[a] For two subcatchments the percentage routed was estimated at 0% and 100% respectively. A single percentage was calibrated and shared by the three remaining subcatchments.

as well as a more detailed model, even though total runoff volumes were similar. Sun et al. (2014) and Krebs (2014) found that a finer discretization resulted in parameter values that were more applicable to other study sites and events. Petrucci and Bonhomme (2014) found that using additional geographic information to increase the spatial resolution could improve model performance, since some model parameters can then be estimated directly from geographic data (see also Dongquan et al.,

(2009); Warsta et al., (2017). To investigate the impact of calibration data selection on these findings and to check them for a predominantly green urban catchment, two levels of discretization were compared: (1) the basic model set-up (the high-resolution model described in Sect. 2.2), and (2) a simpler, more traditional set-up using five subcatchments. In the latter case, each subcatchment was created by aggregating multiple smaller subcatchments from the high-resolution model. The area and percentage imperviousness of each aggregated subcatchment were calculated from its constituent smaller catchments. The cal-

ibration parameters were modified accordingly, as shown in Table 3, with the total number of calibration parameters (including rainfall multipliers) being the same.

## 2.5   Objective functions

The objective function used for the calibrations was the Nash-Sutcliffe model efficiency:

$$\text{NSE} = 1 - \frac{\frac{1}{n}\sum_{i=1}^{n}(S_i - O_i)^2}{\frac{1}{n}\sum_{i=1}^{n}(O_i - \bar{O})^2} \tag{1}$$

Where O denotes observed values and S simulated values. The NSE measures the variance of the model errors (the numerator) as a fraction of the variance of the observations (the denominator). This fraction is then scaled so that it extends from –infinity (i.e., the worst possible fit) via 0 (the score that would be achieved by using the average of observations) to 1, for a

perfect fit. The NSE is dimensionless, so it allows comparing runoff events of different magnitudes. However, when the variance of the observations is small (e.g. for small runoff events), it can become quite sensitive to small changes in the simulated hydrograph. The NSE was calculated for each individual event and the average used as the calibration objective. For further assessment of the modelled hydrographs, two metrics related to the peak flow and the hydrograph volume were used. The peak flow ratio (PFR) was defined as the ratio of the highest simulated to the highest observed flow rates, regardless of the times when they occurred:

$$\text{PFR} = \frac{\max S_i}{\max O_i} \tag{2}$$

Where values >1 indicate overestimated simulated peak flows and values <1 indicate underestimated simulated peak flows. Finally, the relative volume error (VE) considers total flow volumes throughout the event:

$$\text{VE} = \frac{\sum_{i=1}^{n}(S_i - O_i)}{\frac{1}{n}\sum_{i=1}^{n} S_i} \tag{3}$$

It is positive when the simulated total flow volume exceeds the observed one and vice versa. Note that the above formula is only valid if the observation interval is constant. The peak flow ratio and volume error were used here since peak flow rates and storage volumes are often the targets that drainage systems are designed for.

The quick response of the studied catchment means that low flow rates may cover a significant part of the event. Measurements in this range have relatively high uncertainties and may be considered less relevant than periods with higher flows. Therefore, it should be avoided that low flows dominate the analysis, which was achieved by including only time steps with observed flow rates >1 L s$^{-1}$ in calculating these metrics.

## 3  Results and discussion

### 3.1  Calibration performance

#### 3.1.1  Baseline calibration

The baseline calibration (i.e. using the high resolution model without flow data perturbations) was successful for all calibration scenarios, with average NSE for all events ranging from 0.68 to 0.85 (see Table 4). The lowest NSE corresponded to the two CSs based on the percentage runoff (T6_QV_ppP and T32S_QV_ppP). This result can be attributed to one event (see 2), for which both CSs resulted in simulated hydrographs with low NSE, in spite of a visually good fit of the observed data. In this case, low NSE resulted from a small timing error and from low flow rates in the event, which lead to a low variance of the observations and, therefore, an NSE that is more sensitive to small simulation errors. For the two-stage calibration scenarios, the individual stages also produced successful calibrations (stage 1 NSE 0.70 - 0.87, stage 2 NSE 0.78-0.87), except for the second stage in T32S_QV_ppP for the reasons explained above. The NSE for the individual calibration events in the different

**Table 4.** Calibration results. Bold font indicates the best value in each column.

| | High resolution model | | | | | Low resolution model | | | Mean NSE |
| | Baseline | | | Flow-40% | Flow +40% | | | | |
| | NSE | VE | PFR | NSE | NSE | NSE | VE | PFR | |
|---|---|---|---|---|---|---|---|---|---|
| N_T6 | 0.80 | -0.07 | 0.93 | 0.77 | 0.76 | 0.84 | 0.03 | 0.85 | 0.78 |
| T6_P_sum | 0.75 | -0.11 | 0.96 | 0.65 | 0.65 | 0.75 | -0.07 | 0.90 | 0.68 |
| T6_PI_mean | 0.77 | -0.04 | 0.90 | 0.63 | 0.78 | 0.77 | 0.02 | 0.86 | 0.73 |
| T6_PI_30m | 0.74 | -0.09 | 0.95 | 0.72 | 0.72 | 0.74 | -0.05 | **0.95** | 0.72 |
| T6_Q_max | **0.85** | -0.03 | 0.89 | 0.82 | **0.84** | **0.86** | 0.04 | 0.86 | **0.84** |
| T6_Q_60m | 0.79 | -0.09 | 0.91 | 0.77 | 0.77 | 0.81 | **0.01** | 0.90 | 0.78 |
| T6_QV_ppP | 0.68 | -0.11 | 0.89 | -0.10 | 0.65 | 0.65 | -0.09 | 0.94 | 0.41 |
| T6_D_prec | 0.74 | -0.10 | 0.92 | 0.72 | 0.69 | 0.81 | -0.02 | 0.86 | 0.72 |
| T32S_P_sum | 0.83 | 0.03 | 0.90 | 0.77 | 0.83 | 0.68 | 0.08 | 0.74 | 0.81 |
| T32S_PI_mean | 0.83 | 0.03 | 0.96 | 0.75 | 0.80 | 0.78 | 0.05 | 0.84 | 0.79 |
| T32S_Q_max | 0.82 | 0.06 | 0.86 | 0.79 | **0.84** | 0.80 | 0.07 | 0.78 | 0.81 |
| T32S_Q_60m | 0.79 | 0.04 | **0.98** | 0.73 | 0.76 | 0.73 | 0.02 | 0.93 | 0.76 |
| T32S_QV_ppP | 0.70 | 0.06 | 0.85 | 0.62 | 0.73 | 0.67 | 0.11 | 0.75 | 0.68 |
| T32S_D_prec | 0.76 | **0.02** | 0.97 | **0.83** | 0.73 | 0.84 | 0.03 | 0.85 | 0.77 |

calibration scenarios is similar to that reported by Krebs et al. (2013). Overall, the two scenarios based on peak flow performed best (being the only CSs with mean NSE > 0.8) while the two scenarios based on percentage runoff performed worst (only CSs with mean NSE < 0.7).

For the two-stage calibrations the assumption that no runoff occurred from green areas during the first stage of the calibration was checked. During the actual first-stage calibration (i.e. with green area parameters set to default values) there was no runoff from green areas for any of the calibration events in any of the calibration scenarios, so the first stage calibration attributed all runoff to impervious areas as assumed beforehand. However, some runoff occurred from green areas for first-stage events when the calibrated parameter values from the second stage were applied. This runoff was caused by impervious areas draining to green areas. The runoff from green areas was <5% of the total simulated runoff volume for 4 model runs, <10% for an additional 3 runs, and 11.6%, 11.7%, 21.7%, 22.9% and 25.7% respectively for 5 additional runs. These last 5 runs concerned 3 different events with a percentage runoff (calculated before applying rainfall multipliers) between 11% and 12%. Such events may be expected to include some green area runoff and it could be considered to exclude these from the first stage calibration (not done here to limit the complexity of the procedure as discussed in Sect 2.3). In addition, all three events were also included in other first-stage calibrations where they did not result in any significant simulated green area runoff. Removing these events from the first stage of calibration based on initial calibration results would therefore result in the same event being included

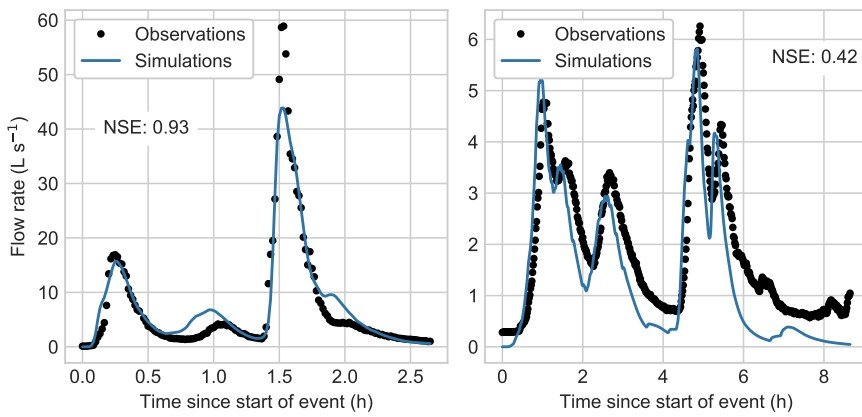

2 example hydrographs run130.pdf

**Figure 2.** Examples of hydrographs for events with high (left) and low (right) objective function (NSE) values.

in different stages for different calibration scenarios, which was considered undesirable. Overall we believe that, although the assumption that all runoff is from directly connected impervious areas when QV_ppP <12% is violated in some cases, the assumption that these events are suitable for calibrating impervious area parameters does hold to a sufficient degree, as also evidenced by the good first-stage calibration performance (see first paragraph of this subsection). In addition, checking for
green area runoff as done here is only possible after calibration, and considering it when selecting events would thus create a more complex, iterative calibration procedure, which would limit the practical applicability of this approach. We considered this to be beyond the paper's original scope of examining different strategies for calibration event selection.

### 3.1.2 Low-resolution model

Calibration runs with a model setup consisting of five instead of 140 subcatchments showed NSE similar to that of the baseline
run (Table 4): the change in performance ranged from +0.08 (T32S_D_prec) to -0.06 (T32S_Q_60m), with only T32S_P_sum showing a larger loss of 0.15. The peak flows predicted by the low-resolution models were most often lower than in the high-resolution model and as a result, peak flow ratios were worse. This effect was stronger for the two-stage calibrations than for the single-stage calibrations. Overall runoff volume was higher in the low-resolution models, which resulted in a smaller volume error. These findings on peak flows and total flow volumes confirm earlier findings by Tscheikner-Gratl et al. (2016).
The changes in peak flow performance were smaller than reported by Krebs et al. (2016), but the changes in NSE and volume errors were comparable.

### 3.1.3 Sensitivity to structural flow measurement errors

Calibration results (NSE) are shown in Table 4 for the cases of structural flow data errors of -40% and +40%. For most calibration scenarios there was a small loss in NSE, except for T6_QV_ppP, which failed to calibrate with an NSE of -0.1
when the flow data was reduced by 40%. Three of the events in that scenario calibrated well (NSE 0.76 - 0.95), but the other

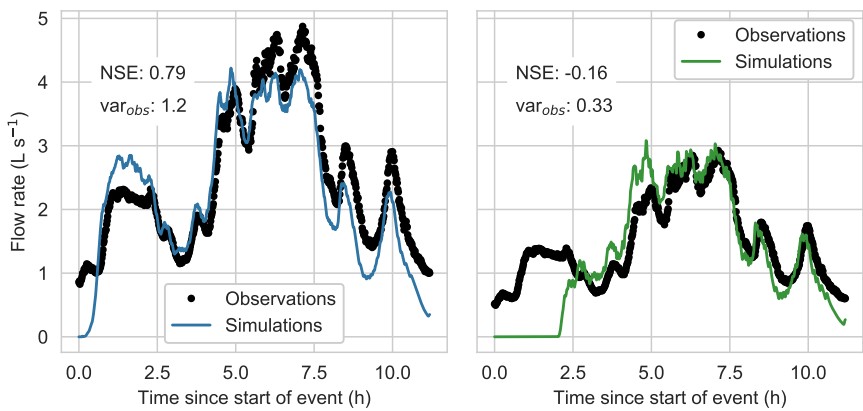

**Figure 3.** Calibrated hydrographs for T6_QV_ppP in the baseline run (left) and after reducing all flow measurements by 40% (right). The low NSE in the right panel is caused by the low variance of the observations.

three produced negative NSE values. These latter three events all missed the first runoff peak; for two of them the quality of fit, judged visually, was the same as in the baseline run, but since the flow rates were low, the variance of the observations was low and thus the NSE values were unsatisfactory (see Figure 3 for an example). T6_PI_mean included one event, for which the reduction of flow observations by 40% resulted in a hydrograph where large parts fell below the 1 L s$^{-1}$ threshold. Except for the events described above, the flow errors could be compensated for in calibration, confirming the earlier findings in the literature (Dotto et al., 2014). In the paper by Dotto et al. the perturbations in flow data resulted in different calibrated values for the percentage imperviousness of the catchment, while in the current paper the perturbations resulted in different values for the rainfall multipliers as discussed in Sect. 3.2.2.

## 3.2 Calibrated parameter values

### 3.2.1 Hydrologic model parameters

Figure 4 shows the calibrated parameter values (for the baseline run), normalized with respect to their calibration ranges (see Table 2). There is considerable variation among the calibrated values obtained in different calibration scenarios, demonstrating that even for parameters with a clear physical interpretation, identification of the best (ideal) value is not straightforward. Gupta et al. (1998) also found considerable variation in the parameter values obtained when using different years as calibration periods for a natural catchment model. Nonetheless, the span of parameter values is considerably reduced compared to the range imposed during calibration, showing that the boundaries were not set too tightly and that the calibration procedure does offer benefits over estimating parameter values directly.

Calibrated parameter values are always uncertain estimates. This uncertainty has been investigated for urban drainage models and shown to be dependent on parameter type, study catchments, model structures, catchment discretization and measurement

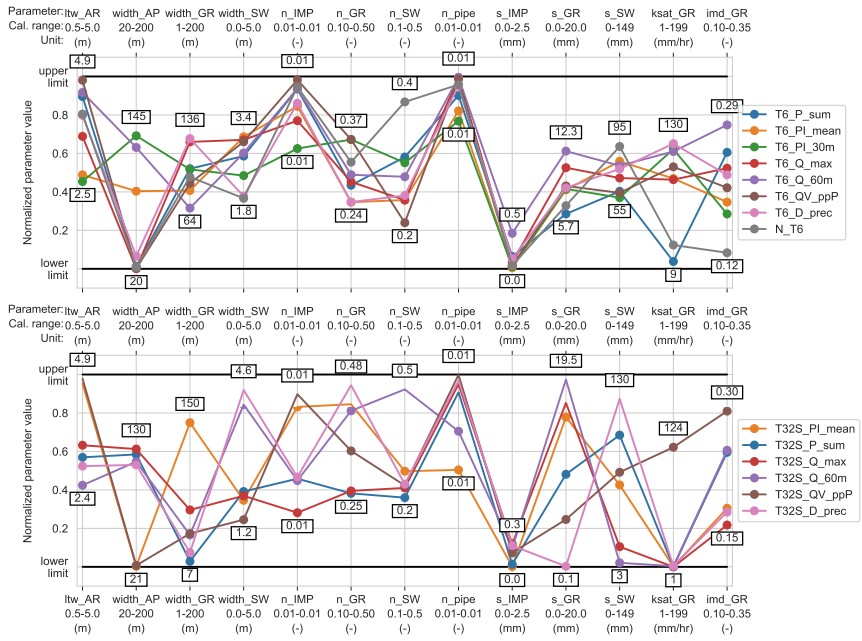

param values.pdf

**Figure 4.** Normalized calibrated parameter values for different calibration scenarios and the baseline run. The highest and lowest values found for each parameter are indicated.

errors (Dotto et al., 2009, 2011, 2014; Kleidorfer et al., 2009a; Sun et al., 2014). The variation found here among the optimum parameter values obtained in different calibration scenarios suggests that the selection of calibration events could also affect the uncertainty of parameter estimates and this influence should be investigated further.

### 3.2.2 Rainfall multipliers

5    The values of rainfall multipliers found in the calibration process ranged from 0.48 to 2.92, showing that there could be significant measurement errors (in precipitation and/or flow) and/or differences between the gauge rainfall and the catchment average rainfall matching best the observed flow rates. For rainfall events that were included in multiple calibration scenarios, the calibrated multipliers from different scenarios were close to each other (see 5). This variation was much smaller than that for the hydrological model parameters (see Sect. 3.2.1). The average value of the rainfall multipliers across all events was 1.2.

10    When all flow data was decreased by 40%, prior to calibration, the different CSs remained in agreement with each other, except for T6_QV_ppP, which failed in this run. The average rainfall multiplier across all events was 0.76 (i.e., 37% lower than in the run without any perturbation of flow data). When all flow data was scaled up by 40%, T32S_P_sum and T32S_Q_max produced deviating multipliers (compared to the other calibration scenarios) for three events each, but the quality of fit was the

**Table 5.** Baseline run calibrated rainfall multipliers for events that were used in at least three CSs.

| Event # | N_T6 | T32S_D_prec | T32S_P_sum | T32S_PI_mean | T32S_Q_60m | T32S_Q_max | T32S_QV_ppP | T6_D_prec | T6_P_sum | T6_PI_30m | T6_PI_mean | T6_Q_60m | T6_Q_max | T6_QV_ppP | Mean | New P | New QV_ppP |
|---|---|---|---|---|---|---|---|---|---|---|---|---|---|---|---|---|---|
| 199 | | | | | | | | 0.58 | 0.58 | | | | | | 0.58 | 8.0 | 21.4 |
| 209 | | | | 0.48 | | | | | | | 0.48 | | | | 0.48 | 3.8 | 14.3[a] |
| 211 | | 0.70 | 0.70 | | 0.70 | 0.70 | | | | | | | | | 0.70 | 6.8 | 15.8[a] |
| 214 | | | | | | 1.16 | | | | | | | | | 1.16 | 7.4 | 8.7 |
| 222 | | | 0.68 | | 0.68 | | | | | | 0.68 | | | | 0.68 | 6.7 | 10.6 |
| 270 | | 1.24 | 1.22 | | | | 1.28 | 1.26 | | | | | | | 1.25 | 11.7 | 9.1 |
| 306 | | | | 0.74 | | | | | | 0.70 | 0.74 | | | | 0.73 | 6.3 | 11.7 |
| 307 | 1.48 | | 1.46 | 1.48 | 1.48 | 1.48 | | | 1.48 | 1.44 | 1.44 | 1.52 | 1.48 | | 1.47 | 44.0 | 11.0[b] |
| 310 | | | 1.06 | 1.06 | | | | | | 1.06 | 1.06 | 1.14 | | | 1.08 | 9.2 | 13.0 |
| 530 | 1.14 | | | 1.10 | 1.10 | 1.12 | 1.04 | | | 1.08 | 1.08 | | 1.14 | | 1.10 | 7.4 | 10.2 |
| 939 | | 0.60 | | | | | | | | | | | | | 0.60 | 4.2 | 9.5 |
| 962 | | | | | | | | | | | | | | 0.98 | 0.98 | 8.3 | 25.4 |
| 971 | | | | | | | 1.08 | | | | | | | | 1.08 | 2.8 | 10.4 |
| 978 | 1.38 | 1.38 | 1.34 | | | 1.34 | | 1.40 | 1.42 | | | 1.36 | 1.38 | | 1.38 | 34.4 | 13.9 |
| 982 | 1.22 | | | 1.20 | | | | | | | 1.26 | 1.22 | 1.26 | | 1.23 | 6.9 | 12.8 |
| 984 | | | | | | 2.02 | 1.94 | | | | | 2.12 | 2.00 | 1.90 | 2.00 | 4.8 | 29.6 |
| 995 | | | | | | | 2.92 | | | | | | | 2.88 | 2.90 | 6.1 | 9.9 [b] |
| 997 | | | | | | | | 1.24 | 1.26 | | | | | | 1.25 | 30.8 | 16.6 |
| 1001 | 1.70 | 1.66 | 1.60 | | | 1.64 | | 1.66 | 1.66 | 1.60 | | 1.64 | 1.70 | 1.64 | 1.65 | 58.2 | 15.1 |
| 1004 | | | | | | | | | | | | | | 0.78 | 0.78 | 3.3 | 32.3 |
| 1019 | 1.46 | 1.48 | | | | | | 1.46 | 1.44 | | | | | | 1.46 | 32.6 | 14.5 |
| 1028 | | | | | | 1.30 | | | | | | | | 1.30 | 1.30 | 3.7 | 33.4 |

[a] Event percentage runoff switches from <12% to >12% when applying rainfall multiplier.

[b] Vice versa.

same across all CSs (according to both the NSE and visual comparison). The average value of the multipliers across all events was 1.59 (i.e., 33% higher than in the baseline run).

The close inter-CS agreement and the similarity in between the magnitude of perturbations in flow data and the magnitude of the corresponding change in rainfall multipliers indicate that the rainfall multipliers work as intended, i.e. compensating for discrepancies between the observed and best-fitting rainfall, rather than for other aspects of catchment runoff modelling. In this respect, the average multiplier of 1.2 in the baseline run suggests that there was some structural disagreement between the observed rainfall and flows.

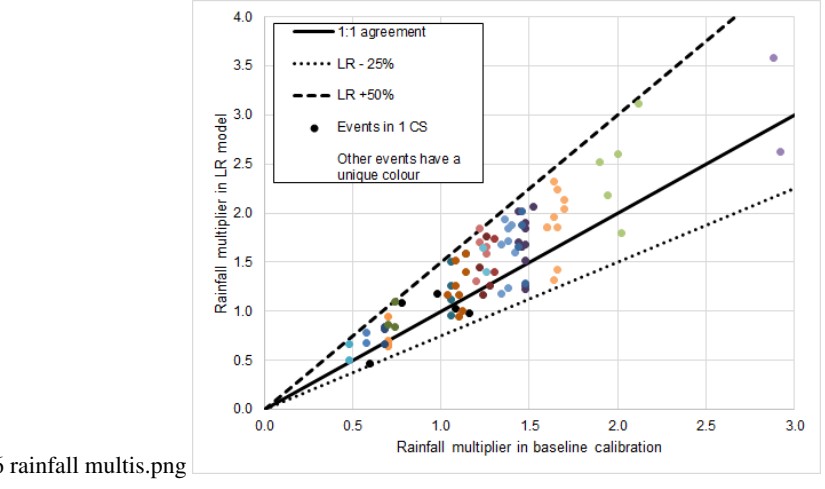

6 rainfall multis.png

**Figure 5.** Rainfall multipliers in baseline calibration (horizontal axis) compared to the LR-model calibration (vertical axis). Each dot is a rainfall multiplier calibrated by one calibration scenario for one event. Identical events appearing in multiple calibration scenarios share the same colour.

In runs with the low-resolution model, contrarily to those with the high-resolution model, there was considerable variation in the values of the rainfall multipliers for each event found by the different calibration scenarios, as shown in Figure 5. The multiplier values obtained ranged from 25% lower to 50% higher, for the same event in the same calibration scenario, compared to the baseline calibration. Three of the low-resolution two-stage calibrations (T32S_D_prec, T32S_Q_60m, T32S_Q_max) found lower multipliers than in the baseline calibration, T32S_QV_ppP had three higher and three lower multipliers and other CSs had all higher multipliers. This behaviour indicates that, in spite of yielding similar results, the rainfall multipliers in the LR-model were used to compensate (within a single event) for the effects of the specific parameter set found in calibration, rather than to compensate for a structural discrepancy between the observed rainfall and flow data as in the baseline calibration (as was the case for the HR models). That the rainfall multipliers appear to behave in a more physical way in the high-resolution model is in line with earlier findings about more transferable parameter values resulting from high-resolution models (Krebs et al., 2014; Sun et al., 2014).

### 3.3 Validation performance

### 3.3.1 Individual events

The successful calibrations predicted 8-13 out of the 19 validation events satisfactorily (NSE >0.5), see Table 6. T6_PI_30m (9 events) and T6_Q_60m (8 events) performed worst while T32S_PI_mean performed best. Perturbations of the flow data in the calibration period led to a lower number of satisfactorily predicted events for most CSs. The two-stage calibration scenarios were less sensitive to perturbations of the flow data in the calibration period, i.e. they predicted more validation events satisfactorily than their single-stage counterparts. When switching from the high resolution to the low-resolution model

**Table 6.** Number of validation events with NSE >0.5 out of 19 total events. Bold font indicates the best value in each column.

|  | Baseline | Cal. flow -40% | Cal. flow +40% | Low-res. model | Total |
| --- | --- | --- | --- | --- | --- |
| N_T6 | 12 | 10 | 8 | 7 | 37 |
| T6_D_prec | 11 | 9 | 9 | 6 | 35 |
| T6_P_sum | 11 | 9 | 9 | 8 | 37 |
| T6_PI_30m | 9 | 9 | 9 | 9 | 36 |
| T6_PI_mean | 10 | 6 | **12** | 6 | 34 |
| T6_Q_60m | 8 | 9 | 9 | 6 | 32 |
| T6_Q_max | 12 | 9 | 11 | 10 | 42 |
| T6_QV_ppP | 12 | 7[a] | 9 | 10 | 31 |
| T32S_D_prec | 12 | **12** | **12** | 10 | 46 |
| T32S_P_sum | 10 | 9 | 10 | **13** | 42 |
| T32S_PI_mean | **13** | **12** | **12** | **13** | **50** |
| T32S_Q_60m | 10 | 9 | 9 | 10 | 38 |
| T32S_Q_max | 11 | 8 | 10 | 12 | 41 |
| T32S_QV_ppP | 11 | **12** | 10 | 12 | 45 |

[a] Run was unsuccessful in calibration

the single-stage CSs were no longer able to predict up to 5 events, while from the two-stage CSs only T32S_D_prec lost two events, and T32S_P_sum, T32S_Q_max, and T32S_QV_ppP actually predicted a higher number of events satisfactorily. Over all four calibration runs, the two-stage calibrations were able to predict more events satisfactorily than their single-stage counterparts.

The events that most often caused failure in validation were four events with peak flow rates of 10 L s$^{-1}$ or less, and therefore, such failures may be attributed to: (1) relatively high measurement uncertainties, and (2) high sensitivity of the NSE to even small changes in the hydrographs. However, it should be noted that the two smallest events (both with a peak flow rate of 4.6 L s$^{-1}$) were predicted with NSE>0.5 by some calibration scenarios. For the other CSs, examination of the hydrographs showed that they predict well the magnitude of events, but produce wrong timing. Another event that failed in validation for all

CSs was that with the highest peak flow rate (53 L s$^{-1}$, see Table A1), which was overestimated by a factor of up to three. This event was dominated by an intense, single-peak burst of rainfall (the highest 30-minute average rainfall intensity was 11.1 mm hr$^{-1}$), so it could have suffered from high spatial variation of the rainfall.

The peak flow ratios obtained for the 19 validation events using the calibrated models from the baseline are shown in the upper panel of Figure 6. Under- or overestimation of peak flows and runoff volumes by the model could lead to an under-

or over-dimensioned system design, and it is therefore relevant to consider these aspects alongside the NSE. Underestimation of peak flows was most frequent, but the largest errors occurred when the flow was overestimated. The variation among CSs was generally larger when the prediction error was larger. The corresponding figure for volume errors is shown in the middle

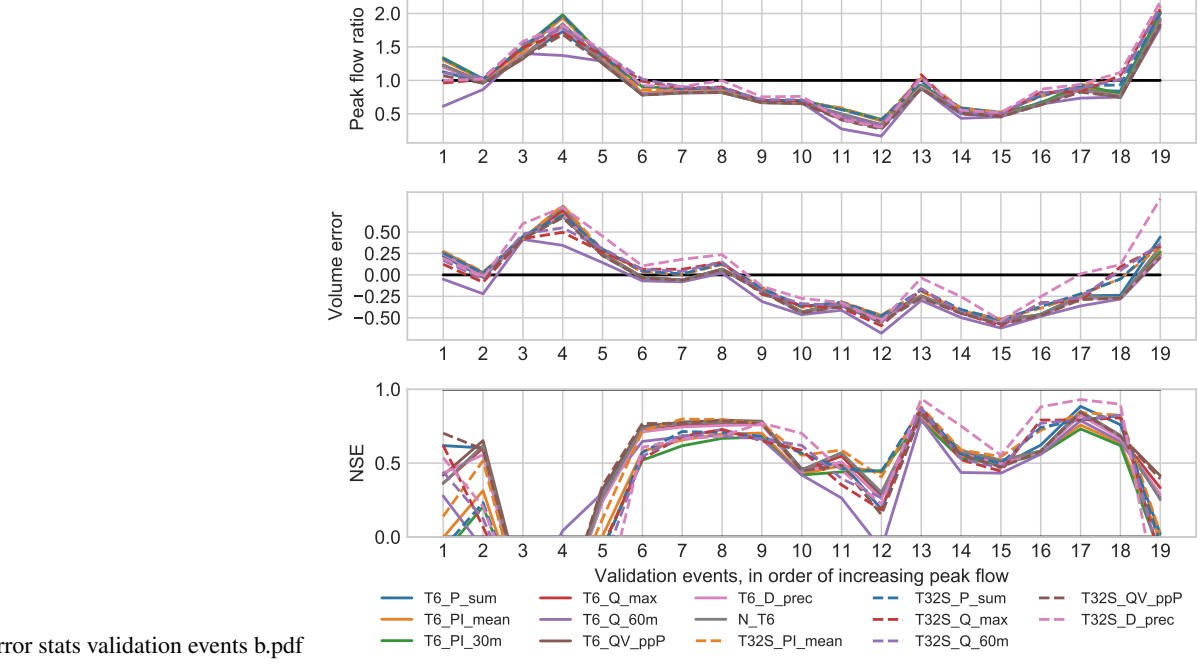

7 error stats validation events b.pdf

**Figure 6.** Error statistics for individual validation events for all calibration scenarios in the baseline runs.

panel of Figure 6. Again, underestimation was more common, but overestimation did occur for a limited number of events. For both peak flows and total volumes, the variation among events was generally larger than the variation among different calibration scenarios, showing that selecting a limited number of validation events may also influence the results of the model evaluation. Across all CSs, two-stage versions had similar or better performance in terms of total runoff volume. Peak flow
ratios were <1 for most events, but for the events that generally did poorly in validation (see above) peak flows (as well as flow volumes) were over predicted instead. The results for both total volumes and peak flows indicate that for most events flows were underestimated, which may be (at least partially) attributed to the discrepancies between observed rainfall and flow found in the calibration phase (see Sect. 3.2.2).

When examining the NSE of the validation events (see the bottom panel of Figure 7), more variation among the different
CSs became visible, although the amount of variation was still event-dependent: inter-CS variation for the same events varies from 0.15 to 1.25. This shows that some events can have a much larger impact on the overall validation results than others. Out of the 19 events, 6 were predicted satisfactorily (NSE>0.5) by some CSs but not by others; 5 events failed for all CSs, and 8 were predicted satisfactorily by all CSs.

### 3.3.2   Overall performance of the high-resolution model

To assess the overall performance of different calibration scenarios for the validation period, several ways of combining the individual events were considered (see Table 7). The simplest metric is obtained by using the NSE means, which ranged from

**Table 7.** Summarized performance for all 19 validation events for the high-resolution model. Bold font indicates the best value in each column.

| | Mean NSE | Clip mean NSE | Median NSE | Joint NSE | # neg NSE | # good NSE | Joint VE | Mean PFR |
|---|---|---|---|---|---|---|---|---|
| N_T6 | 0.33 | 0.45 | 0.58 | 0.65 | 2 | 12 | -0.24 | 0.91 |
| T6_P_sum | 0.39 | 0.45 | 0.60 | 0.66 | 2 | 12 | -0.23 | 0.91 |
| T6_PI_mean | 0.18 | 0.33 | 0.51 | 0.59 | 4 | 10 | -0.24 | 0.96 |
| T6_PI_30m | 0.13 | 0.29 | 0.49 | 0.57 | **5** | 9 | -0.24 | 0.98 |
| T6_Q_max | 0.34 | 0.44 | 0.58 | 0.65 | 2 | 12 | -0.24 | 0.92 |
| T6_Q_60m | 0.37 | 0.37 | 0.43 | 0.60 | 3 | 8 | -0.29 | 0.81 |
| T6_QV_ppP | 0.36 | **0.47** | 0.58 | 0.67 | 2 | 12 | -0.24 | 0.90 |
| T6_D_prec | 0.34 | 0.43 | 0.56 | 0.64 | 2 | 11 | -0.25 | 0.91 |
| T32S_P_sum | 0.19 | 0.34 | 0.56 | 0.68 | **5** | 10 | -0.15 | 0.99 |
| T32S_PI_mean | 0.26 | 0.44 | 0.59 | **0.70** | 2 | **13** | -0.16 | **1.00** |
| T32S_Q_max | 0.31 | 0.34 | 0.53 | 0.67 | 4 | 11 | -0.13 | 0.96 |
| T32S_Q_60m | 0.26 | 0.33 | 0.53 | 0.68 | 4 | 10 | -0.13 | 0.99 |
| T32S_QV_ppP | **0.42** | 0.46 | 0.58 | 0.65 | 2 | 11 | -0.26 | 0.87 |
| T32S_D_prec | 0.22 | 0.34 | **0.61** | **0.70** | 4 | 12 | **-0.02** | 1.01 |

0.13 (T6_PI_30m) to 0.42 (T32S_QV_ppP). There are two conceptual problems with this metric: First, since NSE ranges from negative infinity to plus one, one poorly fitting event can offset multiple well-fitting events. Second, two simulated hydrographs of equally poor fit can have rather different (negative) NSE values, producing different impacts on the overall results, which is not justified by a visual comparison. Therefore, this mean metric is not considered a reliable metric for comparisons, when

poorly fitting events are present. The exclusion of low flow (<10 L s$^{-1}$ peak) events avoids this issue, but does not reward calibration scenarios that do manage to predict these events satisfactorily. Another option is to set all NSE values <-1 to -1 before calculating the mean, which results in NSE ranging from 0.29 to 0.47. Adoption of the median NSEs (insensitive to outliers) lead to a higher range of 0.43 to 0.61, showing that the average or overall validation performance depends more on the outlier events than on typical events. A more commonly used approach is to combine all the events into a single time series

prior to calculating the NSE on the joint time series. This procedure indicated satisfactory performance for all CSs (NSE 0.57 – 0.70). The discussion of various metrics shows that caution is needed when averaging performance over multiple events, as metrics may not reflect the fact that a significant number of events is poorly predicted in all CSs (see Table 6).

   The considerations in the previous paragraph concern the NSE and are not necessarily applicable to other statistics in the same way. The volume error (VE) was included in this study to yield some indication of the overall difference between the

modelled and observed runoff volumes over longer time periods. Therefore, this statistic was summarized over all events

using the joint time-series approach. The volume errors were similar for all high-resolution single-stage calibrated models and showed a general tendency to underestimate flow volumes by 25%. For the two-stage calibrated models volume errors were smaller with underestimation of around 15% (except for T32S_QV_ppP).

### 3.3.3 Overall performance of the low-resolution model

The effect of the low-resolution model depended on the calibration scenario considered, see Table 8. Some scenarios scored better in terms of NSE (gains of up to 0.17 and 3 events predicted with NSE >0.5), while others lost performance by the same metrics (up to 0.24 and 5 events). This is a more less consistent than that found by Krebs et al. (2016), who tested high- and low-resolution models of three catchments and found the high-resolution models to perform better in validation for all three. All but one of the two-stage scenarios predicted more events satisfactorily with the low-resolution model than with the
high-resolution model.

For the single-stage calibration scenarios, the volume errors in the LR were twelve to nineteen percent points higher. The two-stage scenarios showed both worsened performance (T32S_P_sum, T32S_PI_mean) and improved performance (T32S_Q_60m and T32S_Q_max, T32S_QV_ppP). When comparing the hydrographs from the two different model discretizations per event, the high-resolution model usually performed better. However, for the last three CSs mentioned, the
low-resolution performed better compared to the other CSs. For T32S_Q_60m and T32S_Q_max, the low-resolution model predicted the observed hydrographs better for most validation events. These three calibration scenarios were also the only ones where the low-resolution model resulted in lower values for the calibrated rainfall multipliers.

### 3.3.4 Sensitivity to structural flow errors

The introduction of structural flow measurement errors into the calibration data had little effect on performance in the validation
phase. Although there were some changes (compared to the baseline calibration) in the overall NSE values, volume errors and peak flow ratios were almost the same for the baseline and disturbed flow data runs. For T6_D_prec, T6_P_sum, T6_Q_60m, and T6_QV_ppP, runoff started later in the validation event when calibration flow data was increased by 40%, but this had a limited influence on the overall performance metrics (NSE, VE and PFR). Only T6_PI_mean was more sensitive to reducing calibration flow data by 40%. This resulted in lower flows (and therefore better fits) in validation events for the five events that
caused problems for most other CSs (i.e. the four lowest and the single highest peak flow rate(s), see Sect. 3.3.1).

### 3.4 Degradation of performance from calibration to validation

In calibration, the NSE for the different calibration scenarios ranged from 0.68 to 0.85, while in validation it ranged from 0.29 to 0.47. The CSs that did better in calibration lost more performance (measured by NSE) when switching to the validation phase (see Figure 7). The range of performance loss for the different calibration scenarios was larger for the low-resolution
model than for the high-resolution model. For the high resolution model all but one of the two-stage calibrations lost more performance when switching to the validation phase than their single-stage counterparts, whereas for the low-resolution model

**Table 8.** Summarized validation performance (over 19 events) for the low-resolution models. Bold font indicates the best value in each column.

| | Mean NSE | Clip mean NSE | Median NSE | Joint NSE | # neg NSE | # good NSE | Joint VE | Mean PFR | LR visually better than HR (# events) |
|---|---|---|---|---|---|---|---|---|---|
| N_T6 | 0.12 | 0.21 | 0.36 | 0.52 | 5 | 7 | -0.43 | 0.50 | 2 |
| T6_P_sum | 0.05 | 0.22 | 0.42 | 0.57 | 6 | 8 | -0.38 | 0.60 | 3 |
| T6_PI_mean | 0.38 | 0.38 | 0.37 | 0.50 | **0** | 6 | -0.43 | 0.59 | 4 |
| T6_PI_30m | 0.43 | 0.43 | 0.50 | 0.58 | 2 | 9 | -0.34 | 0.74 | 5 |
| T6_Q_max | 0.49 | 0.49 | 0.56 | 0.59 | **0** | 10 | -0.36 | 0.64 | 5 |
| T6_Q_60m | 0.29 | 0.29 | 0.36 | 0.49 | 4 | 6 | -0.46 | 0.49 | 3 |
| T6_QV_ppP | 0.37 | 0.37 | 0.51 | 0.54 | 3 | 10 | -0.40 | 0.66 | 4 |
| T6_D_prec | 0.34 | 0.34 | 0.38 | 0.50 | 4 | 6 | -0.44 | 0.51 | 4 |
| T32S_P_sum | **0.51** | **0.51** | 0.55 | 0.66 | 2 | **13** | -0.27 | 0.60 | 4 |
| T32S_PI_mean | 0.44 | 0.46 | 0.60 | 0.69 | 2 | **13** | -0.22 | 0.80 | 5 |
| T32S_Q_max | 0.05 | 0.33 | 0.64 | 0.70 | 5 | 12 | -0.07 | 1.03 | **12** |
| T32S_Q_60m | 0.13 | 0.28 | 0.52 | 0.66 | 4 | 10 | **-0.04** | **1.02** | 11 |
| T32S_QV_ppP | 0.44 | 0.46 | **0.65** | 0.72 | 2 | 12 | -0.18 | 0.79 | 7 |
| T32S_D_prec | 0.29 | 0.38 | 0.56 | **0.76** | 4 | 10 | -0.05 | 0.86 | 4 |

[a] calculated after setting individual event values <-1 to -1.

all but one of the two stage calibrations had a smaller performance loss. The findings in this section demonstrate that good calibration performance is not necessarily indicative of good validation performance and vice versa, and therefore, whenever feasible, validation should be performed. Previous studies found that high-resolution models lead to more transferable parameter estimates (e.g. less loss of performance when switching to validation, Sun et al. (2014), Krebs et al. (2014)), but in the current study this seems dependent on the calibration data set used. For the two-stage calibrations the low-resolution model usually has less loss in performance than the high resolution model.

## 3.5 Single-stage vs. two-stage calibrations

For those selection criteria, for which both single and two-stage calibrations were performed, the results of the two options can be compared directly (see Figure 8). For the high-resolution model, calibration performance of the two-stage CSs was somewhat better than for the single-stage CSs. By contrast, in the validation phase the NSE was better for the single-stage CSs. However, the volume error and peak flow ratio were better for the two-stage calibrations. For the low-resolution model performance was similar or worse for the two-stage calibrations, but in the validation phase the two-stage calibrations most

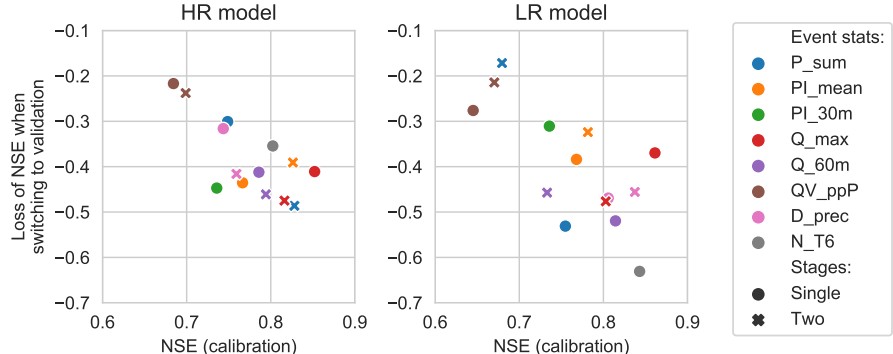

9 v20190710 loss val performance.pdf

**Figure 7.** Loss of performance (NSE) when switching from calibration to validation.

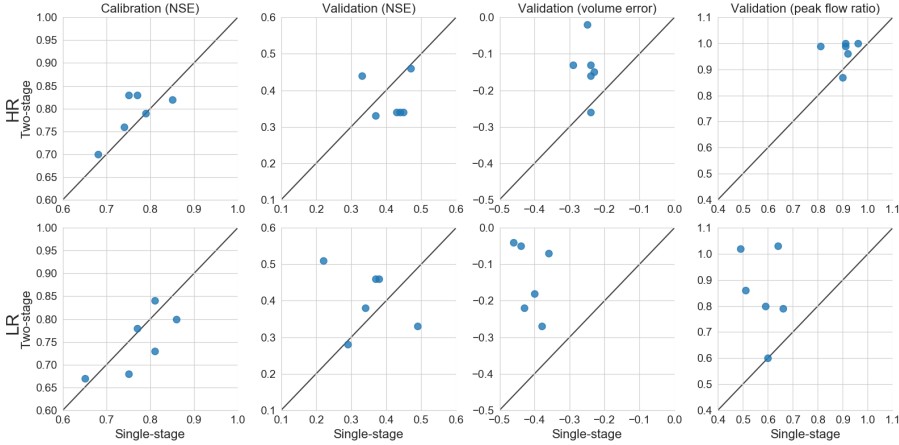

**Figure 8.** Comparison of single-stage and two-stage calibration strategies.

often had higher NSE. In addition, the two-stage calibrations resulted in much better performance in terms of volume error and peak flows than their single-stage counterparts.

## 4   Conclusions

The primary objective of this study was to compare different strategies for the selection of calibration events for a hydrodynamic model of a predominantly green urban area. Two secondary objectives were to verify (1) whether earlier findings on other sources of uncertainty in urban drainage modelling also apply to a greener urban catchment, and (2) whether they are sensitive to the calibration data set used. Calibration strategies consisted of single- and two stage calibrations and considered a number of different metrics by which calibration events can be selected from a larger group of candidate events. Calibration strategies

were tested with high and low spatial resolution models and on data sets with structural flow data errors. The conclusions drawn below are strictly valid for the specific data and catchment characteristics used in this study.

In the baseline run (high resolution model, no structural flow data errors), all calibration scenarios produced successful calibrations (i.e., NSE > 0.5), albeit with varying performance: NSE values ranged from 0.68 to 0.85. For the two-stage calibrations, both stages gave satisfactory results (NSE 0.70-0.87). The two-stage calibrations performed better than their single-stage counterparts in terms of NSE and runoff volume error. The two-stage calibrations also were faster since they reduced the dimensionality (number of simultaneously calibrated parameters) of the calibration problem. Although the obtained values of the SWMM model parameters varied between the different CSs, they found highly similar values for the rainfall multipliers included in the calibration. Switching from a high-resolution to a low-resolution model discretization had only a small impact on calibration performance metrics. However, the values of the rainfall multipliers for each event showed much more variation than with the high-resolution models. Most high-resolution calibration models produced higher values of the multipliers, except for three two-stage CSs, which produced lower values instead. These observations on the rainfall multipliers in low and high-resolution models are in line with previous studies (Krebs et al., 2014; Sun et al., 2014).

The robustness of the calibration scenarios to structural flow errors was tested by calibrating them after uniformly reducing or increasing all flow observations by 40%. Most calibration scenarios were able to adjust to this with only small effects on the calibration performance, except for T6_QV_ppP (six events with highest percentage runoff), which failed in calibration (NSE -0.1) when flow data was reduced by 40%. This can be attributed to two low-flow events, which produced negative NSE values, even though they visually indicated a good fit. This compensation for errors in the calibration data confirms earlier findings from a predominantly impervious catchment(Dotto et al., 2014) for a predominantly green catchment, and confirms that. these findings were insensitive to calibration data selection except in the case of T6_QV_ppP.

The calibrated scenarios were validated against an independent set of 19 validation events. All calibrated scenarios predicted 7 to 13 of these events satisfactorily (NSE >0.5). A group of four events with peak flow rates of less than 10 L s$^{-1}$ caused problems in most calibration scenarios, as did the event with the highest observed peak flow rate. Although most calibration scenarios yielded similar results for the validation events with respect to the overall volume error and the ratio between the modelled and observed peak flow rates, there were considerable differences between the CSs when performance for the validation events was measured by NSE. In terms of NSE the single-stage CSs proved more successful in the validation phase, but for volume error and peak flow error the two-stage CSs performed better. Better performance in regards to flow volumes and peak flows bears more significance for engineering design.

Concerning model discretization, the low-resolution single-stage calibration scenarios showed significantly larger volume errors than their high-resolution counterparts, while most two-stage calibration scenarios showed either the same or even improved volume errors. Two of the two-stage CSs (that also deviated from the others in terms of the calibrated rainfall multipliers) were also the only ones to obtain visually better fitting hydrographs with the low-resolution model setup than with the high resolution model setup. Two-stage calibrations also predicted more validation events satisfactorily when the calibration flow data was perturbed. Earlier studies found that high-resolution models lost less performance when switching to

the validation phase (Krebs et al., 2014; Sun et al., 2014), but, in the current paper, this depended on the set of calibration data that was selected.

*Author contributions.* Ico Broekhuizen maintained the field measurements, validated the data, designed and carried out the simulation experiments, analyzed the results, and drafted the paper. Günther Leonhardt, Jiri Marsalek and Maria Viklander provided feedback on the design of the simulation experiments and reviewed the paper drafts.

*Competing interests.* The authors declare that they have no conflicts of interest.

*Acknowledgements.* This study was funded by the Swedish Research Council Formas (grant number 2015-121). The authors would like to thank CHI/HydroPraxis for providing a license for PCSWMM. The authors would also like to thank Helen Galfi, Ralf Rentz and Karolina Berggren for their work in setting up and maintaining the field measurements.

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

Table A1. Characteristics of all rainfall events used in the validation phase.

| Event # | Precipitation sum in preceding 72 hr | Precipitation sum (P_sum) | Precipitation duration (D_prec) | Average precipitation intensity (PI_mean) | Highest 30-minute average precipitation intensity (PI_30m) | Runoff volume (QV) | Percentage runoff (QV_ppP) | Peak flow rate (Q_max) | Highest 60-minute average flow rate (Q_60m) | Runoff from green areas [a] | Of which originating from imperv. areas [b] | Originating from green areas [c] | Average percentage runoff from green areas [d] |
|---|---|---|---|---|---|---|---|---|---|---|---|---|---|
| | mm | mm | hr | mm hr$^{-1}$ | mm hr$^{-1}$ | mm | % | L s$^{-1}$ | L s$^{-1}$ | mm | mm | mm | % |
| 745 | 0.01 | 10.8 | 26.3 | 0.41 | 3.1 | 1.39 | 12.9 | 10.1 | 5.81 | 0.09 | 0.03 | 0.07 | 0.6 |
| 748 | 0.58 | 3.24 | 11.3 | 0.29 | 2.29 | 0.36 | 11.2 | 28.6 | 6.88 | | | | |
| 757 | 0.33 | 2.02 | 2.57 | 0.79 | 3.38 | 0.13 | 6.34 | 7.28 | 2.52 | | | | |
| 761 | 1.06 | 28.2 | 61.00 | 0.46 | 5.78 | 4.07 | 14.4 | 29.9 | 21.9 | 0.69 | 0.19 | 0.49 | 1.7 |
| 767 | 0.08 | 2.51 | 5.77 | 0.44 | 1.5 | 0.3 | 11.8 | 4.6 | 3.24 | | | | |
| 769 | 0.22 | 2.42 | 2.75 | 0.88 | 2.81 | 0.31 | 12.8 | 16.1 | 6.00 | 0.02 | 0.01 | 0.01 | 0.6 |
| 770 | 2.64 | 6.34 | 7.52 | 0.84 | 8.15 | 0.92 | 14.5 | 45.2 | 16.8 | 0.16 | 0.05 | 0.11 | 1.8 |
| 771 | 8.98 | 3.95 | 4.97 | 0.79 | 4.37 | 0.83 | 21.0 | 30.3 | 15.8 | 0.36 | 0.10 | 0.26 | 6.5 |
| 772 | 12.7 | 17.8 | 20.3 | 0.88 | 5.84 | 3.57 | 20.1 | 35.7 | 26.7 | 1.44 | 0.41 | 1.03 | 5.8 |
| 773 | 21.7 | 8.78 | 8.77 | 1.00 | 3.35 | 1.89 | 21.6 | 17.5 | 11.3 | 0.84 | 0.24 | 0.60 | 6.8 |
| 775 | 26.8 | 5.10 | 14.2 | 0.36 | 3.25 | 1.35 | 26.4 | 32.4 | 10.7 | 0.74 | 0.21 | 0.53 | 10.3 |
| 781 | 0.30 | 6.34 | 11.1 | 0.57 | 2.43 | 0.88 | 13.9 | 23.4 | 6.06 | 0.12 | 0.03 | 0.09 | 1.4 |
| 791 | 0.91 | 9.48 | 13.7 | 0.69 | 11.1 | 0.72 | 7.59 | 53.3 | 13.5 | | | | |
| 793 | 0.01 | 4.97 | 7.08 | 0.70 | 1.86 | 0.32 | 6.37 | 5.60 | 2.70 | | | | |
| 795 | 3.43 | 9.72 | 21.4 | 0.45 | 3.27 | 0.88 | 9.05 | 15.2 | 7.53 | | | | |
| 798 | 9.83 | 2.05 | 5.72 | 0.36 | 1.64 | 0.15 | 7.41 | 4.58 | 2.44 | | | | |
| 799 | 2.13 | 11.4 | 15.9 | 0.72 | 2.55 | 1.20 | 10.6 | 11.1 | 6.24 | | | | |
| 820 | 0.26 | 10.9 | 14.6 | 0.74 | 2.44 | 1.19 | 11.0 | 12.3 | 8.76 | | | | |
| 822 | 11.2 | 20.3 | 17.4 | 1.17 | 6.24 | 3.41 | 16.8 | 51.3 | 28.6 | 0.97 | 0.28 | 0.70 | 3.4 |

[a] Calculated assuming 100% runoff from impervious areas: a = QV – 0.12 P_sum, where 0.12 is the percentage of directly connected impervious area. (Some of this runoff originated from impervious areas that drained to green areas).

[b] Calculated as b = a (25 / (25+63)), where 25 and 63 are the percentages of indirectly connected impervious surfaces and green surfaces respectively.

[c] Calculated as c = a – b

[d] Calculated as d = c / P_sum