# Peer review of "Selection of calibration events for modelling green urban drainage"

_Hydrology and Earth System Sciences, 2019_

## Referee Comment (RC1) · Anonymous Referee #1 · 7 Apr 2019

General Comments: The Authors propose a suitable procedure for selecting calibration events of a hydrodynamic model of a predominantly green urban catchment. A two-stage calibration procedure is used for calibrated first the parameters related to impervious areas, using a set of rainfall events, followed by the pervious area parameters using another set of rainfall events. The selection of calibration events was carried out based on some characteristics such as precipitation intensity, runoff flow rate, flow volume, flow volume as percentage of rain and precipitation duration. The overall ranking of the different calibration scenarios in the validation period is estimated using the statistics of both NSE (Nash-Sutcliffe Efficiency) and RMSE (Root Mean Square Error). The paper address scientific questions within the scope of HESS even if it does not present new concepts or ideas but a rather useful procedure. The scientific methods and assumptions are clearly outlined and the overall presentation is well structured and clear.

Specific Comments: While the calibration strategies (single- and two stage) was already presented by the Authors in a previous paper (1), the different metrics for selecting calibration events from a larger group of candidate events is rather innovative and well described. The risk of using rainfall multipliers is to attribute to the rainfall all the errors due to an incorrect estimate of the model parameters as well as of the model itself. The Authors indicate that the rainfall multipliers compensate for discrepancies between the observed and best-fitting rainfall, rather than for other aspects of catchment runoff modelling by using the baseline model but it is not clear how they reach this conclusion. It should however be considered that rainfall multipliers tend to treat the spatial variability of rain, which has a dynamic effect on the outflow, through a positive or negative variation of rainfall considered uniform on the single watershed and therefore treated in a static way. Figure 5 is not clear and should be conceived in a new way.

Technical Corrections: Table 8 does not contain bold characters as indicated in the text

(1) Ico Broekhuizen I., Leonhardt G., Marsalek J., and Viklander M. (2019). Selection of Calibration Events for Modelling Green Urban Drainage. In: New Trends in Urban Drainage Modelling: UDM 2018 / [ed] Giorgio Mannina, Cham: Springer, 2019, p. 608-613

---

## Referee Comment (RC2) · Anonymous Referee #2 · 18 Apr 2019

General comments

This manuscript presents an analysis of the impact of selecting different sets of calibration data for the SWMM urban hydrological model. Selection is based on a variety of hydro-meteorological characteristics of the available storm events. In addition, the calibration is performed either adjusting all calibration parameters simultaneously, or at two stages where parameters related to pervious and impervious areas are calibrated separately. Finally, the results are analyzed against a backdrop of other sources of uncertainty besides the calibration dataset.

The idea of calibrating impervious area parameters separately using such data where the role of previous areas is presumably insignificant is promising, and in my opinion the results related to this represent the most valuable contribution of the present

manuscript. On the other hand, I struggle to find a novel scientific contribution in the analysis of the calibration event selection in combination with other causes of uncertainty. As argued in the specific comments below the results are inconclusive and it is hard to find any other take-home message than the fact that selection of calibration data has an impact on model parameter values and model performance. This has been established already in existing hydrological literature, as acknowledged also by the authors themselves.

The readability and the quality of the English language are at a very good level.

Specific comments

Study site and data

It would be useful to show somewhere a brief summary of the storm events (e.g. duration, cumulative rainfall depth, cumulative runoff, peak runoff, runoff percentage). The runoff percentage in particular would be interesting as it is used in selecting events for the two-stage calibration. Also, it would be interesting to see to which extend the permeable areas are activated during more intensive events (i.e. runoff-% > 12%).

Event selection

To me the most promising aspect of this manuscript lies in the idea of calibrating parameters related to pervious and impervious areas separately. It is obvious that with a greater runoff percentage than 12% other than just directly connected areas need to contribute. For events with less than 12% runoff it is not equally evident that ONLY directly connected areas contribute. Still, this is a feasible assumption and probably holds to a sufficient extent. There is ample evidence that in urban setting for small events (directly connected) impervious areas predominantly contribute to stormwater flow and for major events also permeable areas are activated.

A couple of issues require further clarification. Did you check whether in the model any runoff was generated from permeable areas when the runoff-% was below 12%? If it is

argued that no runoff is produced outside of the (directly connected) impervious areas for low runoff-% events it should be checked that the model result is consistent with this assumption. Second, the large range of rainfall multipliers (0.48 – 2.92) can make determining the runoff-% somewhat ambiguous. Presumably, the 12% runoff threshold was based on the measured values of precipitation and discharge before applying the rainfall multipliers. Did it happen that a smaller than the unity rainfall multiplier changed the initially below 12% runoff event to exceed the 12% threshold after rainfall multiplier calibration? If yes, should such an event be included in the first stage calibration?

Other sources of uncertainty

The reasoning in including some of the uncertainty sources while leaving others out is not quite clear to me. Also, the take-home what readers should learn from this exercise should be clarified.

Rainfall input

The authors report that reducing flow measurements by 40% leads to 37% reduction in the mean value of rainfall multipliers, and increasing flow measurements by 40% results in a 33% increase in the rainfall multiplier mean value. This seems like rather a trivial result. A more justified description about the purpose of scaling the discharge by a constant multiplier, which causes a corresponding change in the rainfall depth scaling parameter, is needed.

Calibration data measurement uncertainties

See comment above.

Conceptualization / model discretization

While I agree that SWMM is a well established model for urban drainage I do not think that its applicability to areas clearly dominated by pervious areas is equally evident. Presumably in the SWMM runs of the current manuscript the groundwater module has been turned off and infiltration is based on the Green-Ampt equation with infiltration

continuing with a rate appraoching assymptotically the hydraulic conductivity value. It can be questioned whether this is realistic for longer storm events when the soil becomes more saturated. Transpiration is also not accounted for but evaporation only occurs from the depression storage. I am not suggesting that it would feasible to take into account all aspects related to modelling uncertainty. But in my mind the authors' statement "...it is safe to assume that the SWMM conceptualization is appropriate for urban drainage modelling and there was no need to consider this issue further" is in the context of such a low density urban area questionable and does not constitute a valid argument for making a choice about which uncertainty sources are included/excluded in/from the analysis.

Calibration algorithm

The authors state that SCE-UA "...has been widely applied in hydrological applications with great success, so there was no need to subject it to scrutiny in this paper." While I agree that SCE-UA is a powerful tool with an extensive pool of hydrological modelling applications, it is not a sound, objective argument for leaving it out of study. The authors themselves admit that calibration against RMSE can yield a higher NSE than calibration against NSE itself, indicating that the algorithm does not always converge to the optimum value.

Validation performance

Validation performance should be the main argument for improved calibration strategy. If a calibration strategy leads to improved parameter identifiability this should be visible in better results against independent validation data. The authors state that "the two calibration strategies that performed best in the validation period were two-stage strategies" and "...calibrating impermeable and green area parameters in two separate steps may improve the model performance in the validation period...". I think that currently the results about the validation performance for one-stage and two-stage calibrations are inconclusive. The authors use the sum of ranks from several performance

criteria as a proxy for overall performance. Are the results shown in any Table? If yes, I missed them. Also, I would prefer a more quantitative statistic than a sum of class variables (ranks). As NSE is used as the objective criterion for the baseline calibrations it would be a logical choice also for comparing the validation performance. The authors state in Section 3.5. about the validation performance "In terms of NSE, the single-stage calibrations performed better...". On the other hand the 'NSE joint' criterion, typically used for validation (performance over the entire validation data set), seems to be higher for two-stage strategies in Table 6. It is hard for the reader to find guidance here what would be the preferred calibration strategy.

Recommendation

In its current form the manuscript is not in my mind publishable in HESS. The following major changes would be required:

A more informative description of the hydrometeorological data to allow the readers to understand differences between different calibrations

A better justified reasoning for inclusion/exclusion of different error sources

Most importantly, a clear statement about the scientific novelty value of the manuscript where it becomes obvious what are the new findings over just showing that different calibration data lead to different model parameter values and validation performance

Technical comments

Mostly technical comments. The comment for Figure 4 also relates to the content of the manuscript.

Figure 1. Remove the text below the figure (1 map catchment.png). Increase the font size/figure resolution. The legend is hard to read.

Figure 2. Remove the text below the figure (2 example hydrographs run130.pdf).

Figure 3. Remove the text below the figure (3 VE PFR histograms.pdf). In the figure

[Figure]

caption it is stated peak flow ratios to be on the left whereas in the figure the left panel shows the volume error. Please correct.

Figure 4. Remove the text below the figure.

It is hard to interpret with the given information what is causing the negative NSE for the right panel. Is there a timing difference invisible to the eye? Why does the modelled flow stay at zero for the beginning of the event? Clearly there is rain (left panel), so is the diminished rainfall multiplier and/or increased depression storage value causing all rain falling on the directly connected impervious area to be captured in the depression storage?

Figure 5, 6, 7, 8. Remove the text below the figure.

Table 11. Mistake in the NSE single-stage value for D_prec (0.41)? The corresponding value in Table 6 is 0.43?

---

## Author Comment (AC1) · 1 Jun 2019

We would like to thank the referee for taking the time to provide comments on our manuscript. The several issues raised by the referee are addressed one by one below.

**General comments**

Referee's comment: The Authors propose a suitable procedure for selecting calibration events of a hydrodynamic model of a predominantly green urban catchment. A two-stage calibration procedure is used for calibrated first the parameters related to impervious areas, using a set of rainfall events, followed by the pervious area parameters using another set of rainfall events. The selection of calibration events was carried out based on some characteristics such as precipitation intensity, runoff flow rate, flow

volume, flow volume as percentage of rain and precipitation duration. The overall ranking of the different calibration scenarios in the validation period is estimated using the statistics of both NSE (Nash-Sutcliffe Efficiency) and RMSE (Root Mean Square Error). The paper address scientific questions within the scope of HESS even if it does not present new concepts or ideas but a rather useful procedure. The scientific methods and assumptions are clearly outlined and the overall presentation is well structured and clear.

Authors' response: we thank the referee for their supportive comments on the manuscript. Two points require some clarification: First, we would like to clarify that the model performance in the validation period is also assessed based on the flow volume error and the peak flow ratio, and these two statistics are actually the ones where some of the most notable differences between different calibration scenarios are visible. Second, although the ideas of using different types of calibration events and two-stage calibrations are not new in general (as pointed out by the referee), we believe that these issues have not been addressed in any published urban drainage modelling articles, and papers that come close to or touch upon these issues are often focused on urban drainage systems not containing many green areas. Although urban drainage modelling has commonalities with general hydrologic modelling, there are also some key differences, and so findings from natural catchment modelling should not be assumed to also apply to urban drainage modelling. Furthermore, including green areas in urban drainage modelling includes additional processes / process descriptions, and so findings from predominantly impervious catchments (see previous paragraph) should not be assumed to apply in the same way to greener catchments. This can be clarified in the introduction.

Changes in manuscript: add on page 2, end of line 18: Findings from studies of predominantly impervious catchments (see previous paragraph) should not be assumed to apply in the same way to greener catchments and need to be checked for greener catchments.

**Specific comments**

Referee's comment: While the calibration strategies (single- and two stage) was already presented by the Authors in a previous paper (1), the different metrics for selecting calibration events from a larger group of candidate events is rather innovative and well described.

Authors' response: we thank the referee for their supportive comment.

Changes in manuscript: -

Referee's comment: The risk of using rainfall multipliers is to attribute to the rainfall all the errors due to an incorrect estimate of the model parameters as well as of the model itself. The Authors indicate that the rainfall multipliers compensate for discrepancies between the observed and best-fitting rainfall, rather than for other aspects of catchment runoff modelling by using the baseline model but it is not clear how they reach this conclusion.

Authors' response: two arguments support that the rainfall multipliers appear to work as intended, i.e. to compensate for a mismatch between observed rainfall and the rainfall that fits best with the observed outflow: 1. While there is high variability among the obtained parameter values between different calibration scenarios (CSs), there is little variability among the rainfall multipliers for each event as obtained by different CSs. If the multipliers had the effect of compensating for e.g. reduced runoff volumes caused by higher infiltration (e.g. if the calibration parameter saturated hydraulic conductivity was higher), then it would be expected to see inter-CS variation in the rainfall multipliers more similar to that found for the other model parameters. 2. When rainfall input was perturbed by -40Section 3.2.2 of the manuscript can be reorganized to present these arguments more clearly. We would also like to point out that many studies include the catchment area or imperviousness as a way of adjusting flow volumes. The calibrated

area or imperviousness obtained from this will also be affected by the observed vs. best fitting rainfall mismatch. Since high-quality land cover information and field visits were used for catchment delineation in this study we preferred not to further calibrate the catchment size parameter, so as to maintain its clear physical connection to the real system. It was still thought that a mismatch between observed and best-fitting rainfall could be present. Since the other hydrological model parameters (listed in Table 1 and Table 2 of the manuscript) do not have a large effect on the runoff volume, the rainfall multipliers presented a way of accounting for this mismatch.

Changes in manuscript: Reorganize the text in section 3.2.2 so that arguments 1 and 2 above are clearly identifiable as support for the conclusion on the rainfall multipliers. That the role of rainfall multipliers to adjust overall volumes is sometimes filled by calibrating catchment area is already mentioned in section 2.2 (p. 4 line 34 – p. 5 line 2), but the desired effect of maintaining the connection between physical and model catchment size will be added to this sentence.

Referee's comment: It should however be considered that rainfall multipliers tend to treat the spatial variability of rain, which has a dynamic effect on the outflow, through a positive or negative variation of rainfall considered uniform on the single watershed and therefore treated in a static way.

Authors' response: We are fully aware of this issue, but with only one rain gauge and flow sensor being available (and lacking other information on the spatial variability of the rainfall and/or the effect of moving storms in the study area) there was no feasible alternative to assuming uniform rainfall over the catchment. In this light treating the rainfall error as constant over the catchment seems fitting.

Changes in manuscript: add on page 4, line 30: Rainfall multipliers also do not address the spatial variability of the rainfall, but given the lack of multiple gauges or other information about the spatial variability of rainfall in the catchment no clear alternative

was available.

Referee's comment: Figure 5 is not clear and should be conceived in a new way.

Authors' response: We understand that the figure may be somewhat difficult to interpret, but also like that it contains a lot of information in a small amount of space. Improvements can however be made in the way in which the figure is explained. Better labelling of the different numbers in the figure, addition of units for parameter values, and more explanation in the figure caption can be added. It can also be considered to split the figure into multiple panels, each showing a subset of the calibration scenarios.

Changes in manuscript: new version of figure 5

**Technical corrections**

Referee's comment: Table 8 does not contain bold characters as indicated in the text Authors' response: the bold font was inadvertently left out. Changes in manuscript: bold font will be added in the table to indicate the best value in each column.

---

## Author Response (AR1)

We would like to thank the referee for reviewing our manuscript and providing constructive comments. The several issues raised by the referee are addressed one by one below. In addition to the changes in direct response to the reviewers, there are some small edits in the other parts of the manuscript. For these we refer to the marked-up version of the manuscript contained in this file.

**Comments from referee #1**

**General comments**

**(1) Referee's comment:** The Authors propose a suitable procedure for selecting calibration events of a hydrodynamic model of a predominantly green urban catchment. A two-stage calibration procedure is used for calibrated first the parameters related to impervious areas, using a set of rainfall events, followed by the pervious area parameters using another set of rainfall events. The selection of calibration events was carried out based on some characteristics such as precipitation intensity, runoff flow rate, flow volume, flow volume as percentage of rain and precipitation duration. The overall ranking of the different calibration scenarios in the validation period is estimated using the statistics of both NSE (Nash-Sutcliffe Efficiency) and RMSE (Root Mean Square Error). The paper address scientific questions within the scope of HESS even if it does not present new concepts or ideas but a rather useful procedure. The scientific methods and assumptions are clearly outlined and the overall presentation is well structured and clear.

**Authors' response:** we thank the referee for their generally supportive comments on the manuscript. Two points require some clarification:

First, we would like to clarify that the model performance in the validation period is also assessed on the basis of the flow volume error and the peak flow ratio, and these two statistics are actually the ones, for which some of the most notable differences between different calibration scenarios are visible.

Second, we believe that the suggestion that our paper "…does not present new concepts or ideas…" is open to discussion. In fact, the novelty of the paper consists in developing a calibration / validation procedure for a green urban catchment (i.e., with predominantly pervious areas). This is a unique class of catchments, because such green areas may receive water from both rainfall and runoff from adjacent impervious areas, and the resulting runoff is fed into a hydraulically efficient transport network of storm sewers.  Although the ideas of using different types of calibration events and two-stage calibrations are not new in general (as pointed out by the referee or in our introduction section), we believe that the methodology for selection and execution of such procedures has not been addressed explicitly in any published article on urban drainage modelling, and the articles that may peripherally touch upon these issues are generally focused on urban drainage systems, of which runoff is controlled by impervious areas. Although the modelling of urban drainage has commonalities with general hydrologic modelling, there are also some key differences and even differences from modelling conventional urban catchments (Elliott and Trowsdale, 2007; Fletcher et al., 2013). Consequently, specific findings from modelling natural catchments or modelling conventional urban catchments dominated by directly connected impervious surfaces cannot be assumed to apply to green urban catchments without some caveats. This point can be clarified in the introduction.

**Changes in manuscript:** add on page 2, end of line 18: This second aspect also applies to investigations into other sources of uncertainty in urban drainage modelling, some of which have been investigated before, e.g. input and calibration data uncertainties (Dotto et al., 2014; Kleidorfer et al., 2009a) and spatial model resolution (Krebs et al., 2014; Petrucci and Bonhomme, 2014; Sun et

al., 2014). However, these investigations used predominantly impervious catchments and it is, therefore, unknown to what extent their findings apply to greener urban catchments as well and how sensitive such results are to the calibration data set that was used.

Add to the objective statement at the end of the introduction: Two secondary objectives are to verify: (1) the findings from previous urban drainage modelling studies on a greener (less impervious) catchment, and (2) sensitivity of the earlier findings to the calibration data used.

**Specific comments**

**(2) Referee's comment:** While the calibration strategies (single- and two stage) was already presented by the Authors in a previous paper (1), the different metrics for selecting calibration events from a larger group of candidate events is rather innovative and well described.

**Authors' response:** we thank the referee for their supportive comment.

**Changes in manuscript:**         none needed

**(3) Referee's comment:** The risk of using rainfall multipliers is to attribute to the rainfall all the errors due to an incorrect estimate of the model parameters as well as of the model itself. The Authors indicate that the rainfall multipliers compensate for discrepancies between the observed and best-fitting rainfall, rather than for other aspects of catchment runoff modelling by using the baseline model but it is not clear how they reach this conclusion.

**Authors' response:** two arguments support that the rainfall multipliers appear to work as intended, i.e. to compensate for a mismatch between observed rainfall and the rainfall that fits best with the observed outflow:

1. While there is high variability among the obtained parameter values between different calibration scenarios (CSs), there is little variability among the rainfall multipliers for each event as obtained by different CSs. If the multipliers had the effect of compensating for e.g. reduced runoff volumes caused by higher infiltration (e.g. if the calibration parameter saturated hydraulic conductivity was higher), then it would be expected to see inter-CS variation in the rainfall multipliers more similar to that found for the other model parameters.
2. When rainfall input was perturbed by -40% and +40% the rainfall multipliers changed by -37% and +33% respectively. The similarity shows that the rainfall multipliers are sensitive to mismatches between the observed and best-fitting rainfall volume.

Section 3.2.2 of the manuscript can be reorganized to present these arguments more clearly.

We would also like to point out that many studies include the catchment area or imperviousness (defined here as the 'directly connected' imperviousness) as a way of adjusting flow volumes. The calibrated area or imperviousness obtained from this will also be affected by the observed vs. best fitting rainfall mismatch. Since high-quality land cover information and field visits were used for catchment delineation in this study, we preferred not to further calibrate the catchment size parameter, so as to maintain its clear physical connection to the real system. It was still thought that a mismatch between observed and best-fitting rainfall could be present. Since the other hydrological model parameters (listed in Table 1 and Table 2 of the manuscript) do not have a large effect on the runoff volume, the rainfall multipliers presented a way of accounting for this mismatch.

**Changes in manuscript:** Reorganize the text in section 3.2.2 so that arguments 1 and 2 above are clearly identifiable as support for the conclusion on the rainfall multipliers. That the role of rainfall multipliers to adjust overall volumes is sometimes filled by calibrating catchment area is already mentioned in section 2.2 (p. 4 line 34 – p. 5 line 2), but the desired effect of maintaining the connection between physical and model catchment size will be added to this sentence.

**(4) Referee's comment:** It should however be considered that rainfall multipliers tend to treat the spatial variability of rain, which has a dynamic effect on the outflow, through a positive or negative variation of rainfall considered uniform on the single watershed and therefore treated in a static way.

**Authors' response:** We are fully aware of this issue, but with only one rain gauge and flow sensor being available (and lacking other information on the spatial variability of the rainfall and/or the effect of moving storms in the relatively small study area of 10.2 ha) there was no feasible alternative to assuming uniform rainfall over the catchment. Consequently, treating the rainfall error as being constant over the catchment seems fitting.

**Changes in manuscript:** add on page 4, line 30: Rainfall multipliers also do not address the spatial variability of the rainfall, but given the lack of multiple gauges or other information about the spatial variability of rainfall in the catchment no clear alternative was available.

**(5) Referee's comment:** Figure 5 is not clear and should be conceived in a new way.

**Authors' response:** We understand that the figure may be somewhat difficult to interpret, but also like that it contains a lot of information in a small amount of space. However, improvements can be made in several ways: (i) better explanation of the figure; (ii) better labelling of the different numbers in the figure, addition of units for parameter values, and more explanation in the figure caption can be added; or (iii) splitting the figure into multiple vertically aligned panels, each showing a subset of the calibration scenarios.

**Changes in manuscript:** new version of figure 5

**Technical corrections**

**(6) Referee's comment:** Table 8 does not contain bold characters as indicated in the text

**Authors' response:** the bold font was inadvertently left out.

**Changes in manuscript:** bold font will be added in the table to indicate the best value in each column. This will be done in all tables concerning calibration or validation performance.

We sincerely thank the referee for their extensive comments on the manuscript, which we reply to point-by-point below. The referee's comments have been numbered for easy reference. In addition to the changes in direct response to the reviewers, there are some small edits in the other parts of the manuscript. For these we refer to the marked-up version of the manuscript contained in this file.

**Comments from referee #2**

**General comments**

**(10) Referee's comment:** This manuscript presents an analysis of the impact of selecting different sets of calibration data for the SWMM urban hydrological model. Selection is based on a variety of hydro-meteorological characteristics of the available storm events. In addition, the calibration is performed either adjusting all calibration parameters simultaneously, or at two stages where parameters related to pervious and impervious areas are calibrated separately. Finally, the results are analyzed against a backdrop of other sources of uncertainty besides the calibration dataset.

**Authors' response:** this summarizes well the contents of the manuscript.

**Changes in manuscript:** -

**(11) Referee's comment:** The idea of calibrating impervious area parameters separately using such data where the role of previous areas is presumably insignificant is promising, and in my opinion the results related to this represent the most valuable contribution of the present manuscript. On the other hand, I struggle to find a novel scientific contribution in the analysis of the calibration event selection in combination with other causes of uncertainty. As argued in the specific comments below the results are inconclusive and it is hard to find any other take-home message than the fact that selection of calibration data has an impact on model parameter values and model performance. This has been established already in existing hydrological literature, as acknowledged also by the authors themselves.

**Authors' response:** In general, the novelty of our paper consists in: (i) drawing attention to the calibration/validation issues in green urban catchments, and (ii) proposing a calibration / validation procedure for a green urban catchment (i.e., with predominantly pervious areas). This is a unique class of catchments, because such green areas may receive water from both rainfall and runoff from adjacent impervious areas, and the resulting runoff is fed into a hydraulically efficient transport network of storm sewers. Consequently, specific findings from modelling conventional urban catchments dominated by directly connected impervious surfaces cannot be assumed to apply to green urban catchments without some caveats.

Further documentation of innovative aspect of our work is presented in our response to points 17-21, 24, 28 and a newly added emphasis on innovative aspects of our manuscript.

**Changes in manuscript:** see below (points 17-21, 24, 28)

**(12) Referee's comment:** The readability and the quality of the English language are at a very good level.

**Authors' response:** we thank the referee for their supportive comment.

**Changes in manuscript:** none needed

**Specific comments**

**Study site and data**

**(13) Referee's comment:** It would be useful to show somewhere a brief summary of the storm events (e.g. duration, cumulative rainfall depth, cumulative runoff, peak runoff, runoff percentage). The runoff percentage in particular would be interesting as it is used in selecting events for the two-stage calibration. Also, it would be interesting to see to which extend the permeable areas are activated during more intensive events (i.e. runoff-% > 12%).

**Authors' response:** we agree that a table summarizing rainfall-runoff events could be useful to the reader. The table can also contain a rough estimate of how many mm runoff was generated by the green areas. The extent to which green areas are activated can be estimated in a limited way from the data directly (see last column in table C1 in the supplement).

**Changes in manuscript:** such a table will be added in the methods section, see Table C1 in the supplement.

Table C1: characteristics of all rainfall events used in one or more calibration scenarios.

| Event # | Precipitation sum in preceding 72 hr | Precipitation sum (P_sum) | Precipitation duration (D_prec) | Average precipitation intensity (PI_mean) | Highest 30-minute average precipitation intensity (PI_30m) | Runoff volume (QV) | Percentage runoff (QV_ppP) | Peak flow rate (Q_max) | Highest 60-minute average flow rate (Q_60m) | Runoff from green areas [b] | Of which originating from impervious areas [c] | Originating from green areas [d] | Average percentage runoff from green areas [e] |
|---|---|---|---|---|---|---|---|---|---|---|---|---|---|
| | mm | mm | hr | mm/hr | mm/hr | mm | % | L/s | L/s | mm | mm | mm | % |
| 199 | 2.4 | 13.8 | 41.6 | 0.3 | 4.0 | 1.7 | 12.4 | 4.2 | 3.3 | 0.06 | 0.02 | 0.04 | 0.3 |
| 209 | 0.2 | 8.0 | 9.5 | 0.8 | 2.8 | 0.5 | 6.9 | 4.5 | 2.7 | | | | |
| 211 | 8.3 | 9.7 | 22.8 | 0.4 | 6.9 | 1.1 | 11.1 | 29.2 | 11.1 | | | | |
| 214 | 7.3 | 6.4 | 12.1 | 0.5 | 4.3 | 0.6 | 10.1 | 40.5 | 8.5 | | | | |
| 222 | 1.1 | 9.8 | 12.8 | 0.8 | 7.5 | 0.7 | 7.2 | 26.4 | 13.3 | | | | |
| 270 | 0.0 | 9.3 | 38.5 | 0.2 | 3.5 | 1.1 | 11.3 | 22.9 | 8.7 | | | | |
| 306 | 10.1 | 8.6 | 9.1 | 0.9 | 7.1 | 0.7 | 8.5 | 27.5 | 9.3 | | | | |
| 307 | 18.3 | 29.9 | 37.7 | 0.8 | 8.5 | 4.9 | 16.2 | 71.2 | 42.9 | 1.27 | 0.36 | 0.91 | 3.0 |
| 310 | 12.7 | 8.6 | 10.0 | 0.9 | 7.5 | 1.2 | 14.0 | 37.4 | 17.4 | 0.17 | 0.05 | 0.12 | 1.4 |
| 530 | 13.8 | 6.7 | 2.8 | 2.4 | 7.2 | 0.8 | 11.2 | 58.9 | 13.5 | | | | |
| 939 | 0.6 | 7.0 | 25.6 | 0.3 | 1.0 | 0.4 | 5.7 | 2.1 | 1.8 | | | | |
| 962 | 0.0 | 8.5 | 11.2 | 0.8 | 1.4 | 2.1 | 24.9 | 4.9 | 4.4 | 1.09 | 0.31 | 0.78 | 9.2 |
| 971 | 0.2 | 2.6 | 18.6 | 0.1 | 1.1 | 0.3 | 11.3 | 4.0 | 2.9 | | | | |
| 978 | 12.7 | 25.0 | 65.8 | 0.4 | 5.8 | 4.8 | 19.1 | 64.5 | 16.6 | 1.77 | 0.50 | 1.27 | 5.1 |

| 982 | 0.0 | 5.6 | 3.4 | 1.7 | 7.0 | 0.9 | 15.8 | 49.5 | 17.2 | 0.21 | 0.06 | 0.15 | 2.7 |
|---|---|---|---|---|---|---|---|---|---|---|---|---|---|
| 984 | 13.1 | 2.4 | 6.3 | 0.4 | 4.6 | 1.4 | 59.1 | 71.7 | 14.0 | 1.12 | 0.32 | 0.80 | 33.7 |
| 995 | 4.8 | 2.1 | 8.5 | 0.2 | 1.8 | 0.6 | 28.6 | 32.0 | 9.7 | 0.35 | 0.10 | 0.25 | 11.9 |
| 997 | 2.2 | 24.6 | 49.0 | 0.5 | 2.4 | 5.1 | 20.7 | 15.0 | 6.9 | 2.14 | 0.61 | 1.53 | 6.2 |
| 1001 | 0.0 | 35.3 | 56.6 | 0.6 | 8.6 | 8.8 | 25.0 | 56.5 | 32.5 | 4.58 | 1.30 | 3.28 | 9.3 |
| 1004 | 22.5 | 4.2 | 13.9 | 0.3 | 5.9 | 1.1 | 25.2 | 33.3 | 10.6 | 0.56 | 0.16 | 0.40 | 9.5 |
| 1019 | 0.5 | 22.3 | 49.7 | 0.4 | 2.3 | 4.7 | 21.2 | 12.9 | 9.3 | 2.06 | 0.58 | 1.47 | 6.6 |
| 1028 | 6.2 | 2.8 | 7.0 | 0.4 | 1.3 | 1.2 | 43.5 | 6.3 | 4.2 | 0.89 | 0.25 | 0.64 | 22.5 |
|  |  |  |  |  |  |  |  |  |  |  |  |  |  |

a Calculated assuming 100% runoff from impervious areas: a = QV – 0.12 P_sum, where 0.12 is the percentage of directly connected impervious area. (Some of this runoff originated from impervious areas that drained to green areas).
b Calculated as b = a (25 / (25+63)), where 25 and 63 are the percentages of indirectly connected impervious surfaces and green surfaces respectively.
c Calculated as c = a – b
d Calculated as d = c / P_sum

**Event selection**

**(14) Referee's comment:** To me the most promising aspect of this manuscript lies in the idea of calibrating parameters related to pervious and impervious areas separately. It is obvious that with a greater runoff percentage than 12% other than just directly connected areas need to contribute. For events with less than 12% runoff it is not equally evident that ONLY directly connected areas contribute. Still, this is a feasible assumption and probably holds to a sufficient extent. There is ample evidence that in urban setting for small events (directly connected) impervious areas predominantly contribute to stormwater flow and for major events also permeable areas are activated.

**Authors' response:** it can indeed be the case that green areas contribute some runoff even when the percentage runoff is less than 12%. Even impervious surfaces will not generate 100% runoff, so if runoff is exactly 12% it is reasonable to expect that at least a small part of runoff has come from green areas instead. We agree with the referee that the amount of runoff from green areas is small enough that assuming it zero is a feasible assumption. In any case, it would be difficult to determine by how much the 12% threshold should be lowered to ensure that no green area runoff is included, since this would also depend on the antecedent conditions in the catchment. Given a lack of other measurements (e.g soil moisture, standing water in swales) in the catchment it is not possible to tell the initial wetness of the catchment from measurements. Estimating initial conditions using the model itself would lead to the undesirable situation where the value of the threshold (and therefore potentially the set of events to use) would be different for each model run. A fixed percentage is therefore much more workable and probably of more practical use.

**Changes in manuscript:** add the following sentence in section 2.3 on line 11: "(It is conceivable that there is some contribution of green areas when the percentage runoff is less than 12%, and in that case the threshold should be set at a lower value, but since the amount of green area runoff and the appropriate value of the threshold would be highly dependent on antecedent conditions this was not included here.)"

**(15) Referee's comment:** A couple of issues require further clarification. Did you check whether in the model any runoff was generated from permeable areas when the runoff-% was below 12%? If it

is argued that no runoff is produced outside of the (directly connected) impervious areas for low runoff-% events it should be checked that the model result is consistent with this assumption.

**Authors' response:** there are several items to check here:

First, during the first stage calibration (i.e. with default values for green area parameters) there was no runoff from green areas for any of the calibration events in any of the calibration scenarios, and so the first stage calibration attributed all runoff to impervious areas.

Second, using the calibrated parameter values for both impervious and green areas, there were some first-stage events where some runoff was predicted from green areas:

1. When runoff was disabled from both directly and indirectly connected impervious areas /by setting their depression storage to 1000 mm) there were three calibrated models runs (2 for T32S_D_prec, 1 for T32s_Q_60m) that actually generated some runoff from green areas (i.e. the runoff did not originate on impervious areas draining to green areas), but since this was ≤2% of the total simulated runoff volume this was considered negligible.

2. When runoff was disabled only for directly connected impervious areas, a total of 12 calibrated model runs showed non-zero runoff from green areas. This was <5% of total simulated runoff volume for 4 runs, <10% for an additional 3 runs, and 11.6%, 11.7%, 21.7%, 22.9% and 25.7% respectively for the remaining 5 runs. However, almost all of this was runoff that was generated on impervious areas draining onto green areas (see point 1 above).

   Regarding the last mentioned 5 runs, it should be noted that these concerned 3 different events with a percentage runoff between 11% and 12%. Such events may be expected to include some green area runoff and it could be considered to exclude these from the first stage calibration as discussed in comment #14. In addition, all three events were also included in other first-stage calibrations that did not result in any significant simulated green area runoff (0, 0 and 3.4% of total simulated runoff, respectively). Removing these events from the first stage of calibration based on initial calibration results would therefore result in the same event being included in different stages for different calibration scenarios, which we considered undesirable.

   Overall we believe that, although the assumption that all runoff is from directly connected impervious areas when QV_ppP <12% is violated in some cases, the assumption that these events are suitable for calibrating impervious area parameters does hold to a sufficient degree, as also evidenced by the good first-stage calibration performance (mentioned on p 10, l. 2-3). In addition, checking for green area runoff as done here is only possible after calibration, and taking it into account when selecting events would thus create a more complex, iterative calibration procedure which limits the practical applicability of the approach. We considered this to be beyond the paper's original scope of examining different strategies for calibration event selection. It could however be considered as a potential avenue for further research on multi-stage calibration procedures.

**Changes in manuscript:** add a (shorter) version of our response above to section 3.1.1: For the two-stage calibrations the assumption that no runoff occurred from green areas during the first stage of the calibration was checked. During the actual 5 first-stage calibration (i.e. with green area parameters set to default values) there was no runoff from green areas for any of the calibration events in any of the calibration scenarios, so the first stage calibration attributed all runoff to impervious areas as assumed beforehand. However, some runoff occurred from green areas for first-stage events when the calibrated parameter values from the second stage were applied. This runoff

was caused by impervious areas draining to green areas. The runoff from green areas was <5% of the total simulated runoff volume for 4 model runs, <10% for an additional 3 runs, and 11.6%, 11.7%, 21.7%, 22.9% and 25.7% respectively for 5 additional runs. These last 5 runs concerned 3 different events with a percentage runoff (calculated before applying rainfall multipliers) between 11% and 12%. Such events may be expected to include some green area runoff and it could be considered to exclude these from the first stage calibration (not done here to limit the complexity of the procedure as discussed in Sect 2.3). In addition, all three events were also included in other first-stage calibrations where they did not result in any significant simulated green area runoff. Removing these events from the first stage of calibration based on initial calibration results would therefore result in the same event being included in different stages for different calibration scenarios, which was considered undesirable. Overall we believe that, although the assumption that all runoff is from directly connected impervious areas when QV_ppP <12% is violated in some cases, the assumption that these events are suitable for calibrating impervious area parameters does hold to a sufficient degree, as also evidenced by the good first-stage calibration performance (see first paragraph of this subsection). In addition, checking for green area runoff as done here is only possible after calibration, and considering it when selecting events would thus create a more complex, iterative calibration procedure, which would limit the practical applicability of this approach. We considered this to be beyond the paper's original scope of examining different strategies for calibration event selection.

**(16) Referee's comment:** Second, the large range of rainfall multipliers (0.48 – 2.92) can make determining the runoff-% somewhat ambiguous. Presumably, the 12% runoff threshold was based on the measured values of precipitation and discharge before applying the rainfall multipliers. Did it happen that a smaller than the unity rainfall multiplier changed the initially below 12% runoff event to exceed the 12% threshold after rainfall multiplier calibration? If yes, should such an event be included in the first stage calibration?

**Authors' response:** the 12% runoff threshold was indeed applied directly to the measured values of precipitation and discharge.

There were two events where the rainfall multiplier was less than 1 and reduced rainfall so that the new percentage runoff exceeded 12%. This can also be displayed in an extended version of Table 4 from the manuscript, see Table C2 in the supplement. It is of course possible to exclude such events from their respective stages in the calibration and replace them with another event. Being consistent about considering the percentage runoff as calculated using the calibrated rainfall multipliers would also require the following three adjustments as well:

1. It would have to be applied 'in both directions', i.e. second-stage calibration events where the calibrated multiplier was large enough that runoff % was reduced below 12% would have to be excluded from the second stage. (This was the case for two events. For these events they would first have to be considered as replacement for a first-stage event, and the first stage calibration re-run, before redoing the second stage of the calibration. (Depending on the results from this the whole procedure might have to be repeated as well.)
2. All event characteristics related to rainfall (i.e. P_sum, PI_mean, PI_30m, QV_ppP) would have to be re-calculated and the related CSs determined and run again if the event set changed.
3. Out of the 32 events that were available for use in calibration scenarios, only 22 were actually selected by one or more CSs, so calibrated multipliers are not available for the other

10 events. It would be necessary to somehow obtain a calibrated multiplier value for them too so that they may be reconsidered for use in the calibration.

Although this might improve the overall results of the proposed calibration procedure, it would also increase the complexity and raise several new issues, such as how to obtain a calibrated rainfall multiplier for the 10 events that have not yet been used. We considered this to be beyond the paper's original scope of examining different strategies for calibration event selection. It could however be considered as a potential avenue for further research on multi-stage calibration procedures.

**Changes in manuscript:** clarify that event selections were fixed beforehand and not adjusted based on initial calibration results. Add a short version of the explanation above in section 2.3, page 6, end of line 18: Applying the calibrated rainfall multipliers in the calibration (Sect. 2.2) means that event properties relating to rainfall and percentage runoff will change, and the percentage runoff can change from <12% to >12% and vice versa. Doing this consistently for all events in the calibration procedure would require (1) re-calculating which events should be available in each stage, (2) estimating in some way rainfall multipliers for all events, including those not initially selected by any calibration scenario, (3) re-calculating which events are used in each CS, and (4) repeating the calibration for any CS that has had any of its events changed. Although this might improve the overall results of the proposed calibration procedure, it would also increase the complexity and raise several new issues, such as how to obtain a calibrated rainfall multiplier for the 10 events that were not used in any CS. We considered this to be beyond the paper's original scope of examining different strategies for calibration event selection and proposing a practically useable two-stage calibration procedure.

Table C2: calibrated rainfall multipliers and new percentages runoff.

| Event # | N_T6 | T32S_D_prec | T32S_P_sum | T32S_PI_mean | T32S_Q_60m | T32S_Q_max | T32S_QV_ppP | T6_D_prec | T6_P_sum | T6_PI_30m | T6_PI_mean | T6_Q_60m | T6_Q_max | T6_QV_ppP | Mean | New P | New QV_ppP | Swap stage |
|---|---|---|---|---|---|---|---|---|---|---|---|---|---|---|---|---|---|---|
| 199 | | | | | | | | 0.58 | 0.58 | | | | | | 0.58 | 8.0 | **21.4** | |
| 209 | | | | 0.48 | | | | | | | 0.48 | | | | 0.48 | 3.8 | 14.3 | gray > green |
| 211 | | 0.70 | 0.70 | | 0.70 | 0.70 | | | | | | | | | 0.70 | 6.8 | **15.8** | gray > green |
| 214 | | | | | | 1.16 | | | | | | | | | 1.16 | 7.4 | 8.7 | |
| 222 | | | 0.68 | | 0.68 | | | | | | 0.68 | | | | 0.68 | 6.7 | 10.6 | |
| 270 | | 1.24 | 1.22 | | | | 1.28 | 1.26 | | | | | | | 1.25 | 11.7 | 9.1 | |
| 306 | | | | 0.74 | | | | | | 0.70 | 0.74 | | | | 0.73 | 6.3 | 11.7 | |
| 307 | 1.48 | | **1.46** | 1.48 | 1.48 | 1.48 | | | 1.48 | 1.44 | 1.44 | 1.52 | 1.48 | | 1.47 | 44.0 | 11.0 | green > gray |
| 310 | | | | 1.06 | 1.06 | | | | | 1.06 | 1.06 | 1.14 | | | 1.08 | 9.2 | 13.0 | |
| 530 | 1.14 | | | 1.10 | 1.10 | 1.12 | 1.04 | | | 1.08 | 1.08 | | 1.14 | | 1.10 | 7.4 | 10.2 | |

| ID | 1 | 2 | 3 | 4 | 5 | 6 | 7 | 8 | 9 | 10 | 11 | 12 | 13 | 14 | 15 | 16 | 17 |
|---|---|---|---|---|---|---|---|---|---|---|---|---|---|---|---|---|---|
| 939 | | 0.60 | | | | | | | | | | | | 0.60 | 4.2 | 9.5 | |
| 962 | | | | | | | | | | | | | 0.98 | 0.98 | 8.3 | **25.4** | |
| 971 | | | | | | 1.08 | | | | | | | | 1.08 | 2.8 | 10.4 | |
| 978 | 1.38 | 1.38 | **1.34** | | 1.34 | | 1.40 | 1.42 | | | 1.36 | 1.38 | | 1.38 | 34.4 | 13.9 | |
| 982 | 1.22 | | | 1.20 | | | | | | | 1.26 | 1.22 | 1.26 | 1.23 | 6.9 | 12.8 | |
| 984 | | | | | 2.02 | 1.94 | | | | | 2.12 | 2.00 | 1.90 | 2.00 | 4.8 | **29.6** | |
| 995 | | | | | | 2.92 | | | | | | | 2.88 | 2.90 | 6.1 | 9.9 | green > gray |
| 997 | | | | | | | 1.24 | 1.26 | | | | | | 1.25 | 30.8 | 16.6 | |
| 1001 | 1.70 | 1.66 | **1.60** | | 1.64 | | 1.66 | 1.66 | 1.60 | | 1.64 | 1.70 | 1.64 | 1.65 | 58.2 | 15.1 | |
| 1004 | | | | | | | | | | | | | 0.78 | 0.78 | 3.3 | **32.3** | |
| 1019 | 1.46 | 1.48 | | | | | 1.46 | 1.44 | | | | | | 1.46 | 32.6 | 14.5 | |
| 1028 | | | | | | 1.30 | | | | | | | 1.30 | 1.30 | 3.7 | **33.4** | |

**Other sources of uncertainty**

**(17) Referee's comment:** The reasoning in including some of the uncertainty sources while leaving others out is not quite clear to me. Also, the take-home what readers should learn from this exercise should be clarified.

**Authors' response:** some of the issues described have been investigated before for urban drainage models (e.g. data uncertainties by Kleidorfer et al. (2009) and Dotto et al. (2014), model resolution by e.g. Krebs et al. (2014), Petrucci and Bonhomme (2014), Sun et al. (2014) and Tscheikner-Gratl et al. (2016)). The idea behind including other sources of uncertainty was (primarily) to see if different calibration event sets showed different sensitivity to these issues and (secondarily) to see if the findings also applied to a different data set and catchment (more dominated by green areas).

Although we considered it an interesting experiment at the time, the impact of what objective function is used in calibration of urban drainage models has not been investigated extensively before (Barco et al. (2008) made some short remarks), so we would remove this from the manuscript. (A thorough investigation of this would be an interesting topic for a different study.) However the different objective functions used for validation phase (e.g. volume error, peak flow) would still be included since they provide additional insight into the simulation results.

Removing the parts on objective function would also allow to describe in more detail the effect of the model resolution, since it is an interesting finding that some of the benefits of the two-stage calibration (better flow volume and peak flow in validation phase) are stronger for the low-resolution model.

The take-home messages from this are:

1. The impact of perturbed calibration data appears small (confirming the findings by Dotto et al. (2014)), but we do see interaction between the calibration data selection and the model discretization.
2. The two-stage calibration gives better results in terms of flow volume and peak flow in the validation phase, and this effect is much stronger for the low-resolution models.

**Changes in manuscript:**

1. Section 2.4 (other sources of uncertainty):
   a. Describe the aim of including other sources: i.e. check if earlier findings are sensitive to different calibration data sets and if they also apply for a different data set and a greener catchment.
   b. Add references to previous studies on rainfall input uncertainty effect on urban drainage modelling in lines 10-12 (Dotto et al., 2014; Kleidorfer et al., 2009).
   c. Lines 20-24: describe a bit more the Dotto and Kleidorfer papers that are referred to, including that they used more pipe-based drainage systems and a fixed set of events.
   d. Lines 28 and further: add references to articles dealing with model resolution (Krebs et al., 2014; Petrucci and Bonhomme, 2014; Sun et al., 2014; Tscheikner-Gratl et al., 2016)
2. Remove the parts that deal with the calibration using RMSE as alternative objective function:
   a. page 7 lines 25-27
   b. page 9 lines 2-5
   c. Section 3.1.2, including table 3.
   d. Table 5 column 3: "RMSE as obj. func." + update column "total"
   e. Section 3.3.3, including table 7 and figure 8.
   f. Conclusion page 24 lines 10-11
   g. mention in abstract

**Rainfall input**

**(18) Referee's comment:** The authors report that reducing flow measurements by 40% leads to 37% reduction in the mean value of rainfall multipliers, and increasing flow measurements by 40% results in a 33% increase in the rainfall multiplier mean value. This seems like rather a trivial result. A more justified description about the purpose of scaling the discharge by a constant multiplier, which causes a corresponding change in the rainfall depth scaling parameter, is needed.

**Authors' response:** since flow data is obviously an important part in the calibration process, we wanted to see if earlier results from Dotto (2014) and Kleidorfer (2009) would be sensitive to different sets of calibration events. Our findings mainly confirm their work. For urban catchments these issues have only been investigated to a limited extent (i.e. with a single set of events and for rather impervious catchments) so additional support of earlier findings is useful. Other disturbances of / errors in the calibration are conceivable, but were deemed beyond the scope of this study.

In addition, the correlation between the adjustment in rainfall and the adjustment in rainfall multipliers also supports the idea that the rainfall multipliers are compensating (even in the baseline run) for a mismatch between observed and best-fitting rainfall (as discussed in section 3.2.2), and therefore that they are a suitable way of accounting for this mismatch.

A better description of why this aspect is considered is also addressed in our response to comment 17 above.

**Changes in manuscript:** see above.

**Calibration data measurement uncertainties**

**(19) Referee's comment:** See comment above.

**Authors' response:** See response above.

**Changes in manuscript:** See response above.

**Conceptualization / model discretization**

**(20) Referee's comment:** While I agree that SWMM is a well established model for urban drainage I do not think that its applicability to areas clearly dominated by pervious areas is equally evident. Presumably in the SWMM runs of the current manuscript the groundwater module has been turned off and infiltration is based on the Green-Ampt equation with infiltration continuing with a rate appraoching assymptotically the hydraulic conductivity value. It can be questioned whether this is realistic for longer storm events when the soil becomes more saturated. Transpiration is also not accounted for but evaporation only occurs from the depression storage. I am not suggesting that it would feasible to take into account all aspects related to modelling uncertainty. But in my mind the authors' statement "… it is safe to assume that the SWMM conceptualization is appropriate for urban drainage modelling and there was no need to consider this issue further" is in the context of such a low density urban area questionable and does not constitute a valid argument for making a choice about which uncertainty sources are included/excluded in/from the analysis.

**Authors' response:** we appreciate the distinction between the application of SWMM to pervious and impervious areas. It is correct that the groundwater module in SWMM was not utilized in this study, and that therefore only the Green-Ampt equation + drying is used to account for infiltration. As pointed out by the referee, recovery of the infiltration capacity is not based on evapotranspiration, but is instead based on the soil's saturated hydraulic conductivity (Rossman and Huber, 2016).

Our original formulation was perhaps too optimistic, but we still believe that it is reasonable not to treat model structure as an uncertainty source in this article for the following reasons:

1. Unlike input and calibration data and model resolution, model structure uncertainty has not been addressed extensively in the urban drainage modelling literature.
2. There is a lack of methods for considering model structure uncertainty other than using different models, which is outside the scope of this study. The catchment and the high-resolution model also require certain features (e.g. routing runoff from one subcatchment to another subcatchment, support for automated runs) that are present in SWMM and not in other models. Model runtime is also a limiting factor.
3. The Green-Ampt method itself has been in use for many years. Other infiltration models are available (e.g. Horton or SCS curve number in SWMM) but going into these would be outside the scope of this study. Ideally a study on infiltration models in urban drainage modelling would also make use of infiltration and/or soil moisture measurements which are not available here.

**Changes in manuscript:** Replace p8, lines 5-7 with: Although model structure is also a recognized source of uncertainty (Deletic et al., 2012), it was not considered here since (a) there is a lack of previous research on this topic for urban drainage modelling that could be referred to and (b) there is a lack of methods to address this other than using different models in parallel, which was considered outside the scope of this study, and would in any case be difficult since the catchment model requires some SWMM features (e.g. routing runoff from one subcatchment to another, good support for automated runs) which are not always present in other models.

**Calibration algorithm**

**(21) Referee's comment:** The authors state that SCE-UA "… has been widely applied in hydrological applications with great success, so there was no need to subject it to scrutiny in this paper." While I agree that SCE-UA is a powerful tool with an extensive pool of hydrological modelling applications, it is not a sound, objective argument for leaving it out of study. The authors themselves admit that calibration against RMSE can yield a higher NSE than calibration against NSE itself, indicating that the algorithm does not always converge to the optimum value.

**Authors' response:** in relation to the improved description of why different sources of uncertainty are included it's good to mention that (like for objective functions) there is a lack of studies examining the effect of calibration algorithms on urban drainage modelling (Deletic et al., 2012). (And even to some extent in general hydrology (Houska et al., 2015)). A thorough examination of the effect of the calibration algorithm would require implementing many different algorithms. Since there is a lack of earlier studies here to refer to it is acceptable to leave the calibration algorithm out altogether.

**Changes in manuscript:** the statement on the exclusion of the calibration algorithm as source of uncertainty (page 8, lines 4-7) will be rephrased: Likewise, the calibration algorithm (Deletic et al., 2012; Houska et al., 2015) and numerical issues (Deletic et al., 2012; Kavetski 5 et al., 2006) are recognized as sources of uncertainty, but there is a lack of studies addressing these specifically for urban drainage modelling that could be referred to here. Since breaking new ground in these areas was considered beyond the scope of this paper, these sources of uncertainty are not considered here.

**Validation performance**

**(22) Referee's comment:** Validation performance should be the main argument for improved calibration strategy. If a calibration strategy leads to improved parameter identifiability this should be visible in better results against independent validation data. The authors state that "the two calibration strategies that performed best in the validation period were two-stage strategies" and "… calibrating impermeable and green area parameters in two separate steps may improve the model performance in the validation period…". I think that currently the results about the validation performance for one-stage and two-stage calibrations are inconclusive. The authors use the sum of ranks from several performance criteria as a proxy for overall performance. Are the results shown in any Table? If yes, I missed them.

**Authors' response:** the overall ranking is shown for the calibration phase in Table 3, and shown for both calibration and validation phase in Table 9.

Presentation of validation results could be presented better by having all results relating to the baseline calibration (HR model) in one table, i.e. combining tables 6, 9 and 11. A similar table could also be made for all results with the low-resolution model (i.e. replacing table 8) to better illustrate the benefits that the two-stage calibration offers there in terms of flow volume and peak flow performance in the validation phase, since these are more pronounced for the low-resolution model. The results from the low-resolution model were not described extensively in the manuscript but we believe they can strengthen a take-home message for the readers and should therefore be included in more detail. The results are currently best illustrated by Table 4.2 in the corresponding author's licentiate thesis (Broekhuizen, 2019), so the re-organizing of the tables with validation results should also include the data shown there. The table is included below (Table C3 in the supplement) for easy reference, but the data will be organized differently (i.e. one table for the HR model and one for the LR model) in the paper.

**Changes in manuscript:** re-organize tables as described above, and describe in more detail in the text the effects of the single- and two-stage calibrations for the low-resolution model.

**(23) Referee's comment:** Also, I would prefer a more quantitative statistic than a sum of class variables (ranks). As NSE is used as the objective criterion for the baseline calibrations it would be a logical choice also for comparing the validation performance.

**Authors' response:** the validation NSE is already presented in Table 9 to allow for comparison of the different CSs. The problem with any single validation characteristic is that it would either ignore some aspects of model performance or it would have to combine different statistics (i.e. NSE, volume error, peak flow error) in some arbitrary way. E.g. using only the NSE for validation performance would ignore that two-stage strategies perform better in terms of total flow volume and peak flow. We think it is interesting that different statistics give a different view of which calibration strategies perform better and this should be reflected in the manuscript.

**Changes in manuscript:** The discussion will be focused more on discussing the individual performance statistics to highlight that different criteria give a different picture of the effects of calibration data selection.

**(24) Referee's comment:** The authors state in Section 3.5. about the validation performance "In terms of NSE, the single-stage calibrations performed better…". On the other hand the 'NSE joint' criterion, typically used for validation (performance over the entire validation data set), seems to be higher for two-stage strategies in Table 6. It is hard for the reader to find guidance here what would be the preferred calibration strategy.

**Authors' response:** although the single-stage performs better in terms of mean NSE (i.e. NSE calculated for each event, then averaged), it performs worse in terms of joint NSE (i.e. all events collated into a single time series for which NSE is then calculated), joint volume error and mean peak flow ratio. As discussed in section 3.3.2 the downside of the joint NSE is that it can give good scores even when several events are poorly predicted. Therefore joint NSE may be considered too optimistic which is why we did not use it extensively in this paper.

In terms of a take-home message it is important to point out that the two-stage calibration is much faster since it reduces the dimensionality of the calibration problem compared to the single-stage calibration. In addition to this it has sometimes slightly poorer validation performance in terms of NSE but typically better performance according to other characteristics.

The take-home message can also be strengthened by highlighting more the differences between HR and LR models (or rather that the benefits of the two-stage calibration are stronger for the LR model). This is currently best illustrated by Table 4.2 in the corresponding author's licentiate thesis (Broekhuizen, 2019), so the re-organizing of the tables with validation results (see comment #22) should also include the data shown there. The table is included in the supplement as table C3 for easy reference, but the data will be organized differently (i.e. one table for the HR model and one for the LR model) in the paper:

**Table C3**: Calibration and validation performance of single and two-stage calibration scenarios. HR denotes the high-resolution model, LR the low resolution model. The names of the calibration scenarios are explained in paper III.

| Calibration scenario | Calibration (6 events) | | Validation (19 events) | | | | | | | |
|---|---|---|---|---|---|---|---|---|---|---|
| | NSE | | # events NSE > 0.5 | | Mean NSE [a] | | Volume error | | Peak flow ratio | |
| | HR | LR | HR | LR | HR | LR | HR | LR | HR | LR |
| N_T6 | 0.80 | 0.84 | 12 | 7 | 0.45 | 0.21 | –0.24 | –0.43 | 0.91 | 0.50 |
| T6_D_prec | 0.74 | 0.81 | 11 | 6 | 0.43 | 0.34 | –0.25 | –0.44 | 0.91 | 0.51 |
| T6_P_sum | 0.75 | 0.75 | 11 | 8 | 0.45 | 0.22 | –0.23 | –0.38 | 0.91 | 0.60 |
| T6_PI_30m | 0.74 | 0.74 | 9 | 9 | 0.29 | 0.43 | –0.24 | –0.34 | 0.98 | 0.74 |
| T6_PI_mean | 0.77 | 0.77 | 10 | 6 | 0.33 | 0.38 | –0.24 | –0.43 | 0.96 | 0.59 |
| T6_Q_60m | 0.79 | 0.81 | 8 | 6 | 0.37 | 0.29 | –0.29 | –0.46 | 0.81 | 0.49 |
| T6_Q_max | 0.85 | 0.86 | 12 | 10 | 0.44 | 0.49 | –0.24 | –0.36 | 0.92 | 0.64 |
| T6_QV_ppP | 0.68 | 0.65 | 12 | 10 | 0.47 | 0.37 | –0.24 | –0.40 | 0.90 | 0.66 |
| | | | | | | | | | | |
| T32S_D_prec | 0.76 | 0.84 | 12 | 10 | 0.34 | 0.38 | –0.02 | –0.05 | 1.00 | 0.86 |
| T32S_P_sum | 0.83 | 0.68 | 10 | 13 | 0.34 | 0.51 | –0.15 | –0.27 | 0.99 | 0.60 |
| T32S_PI_mean | 0.83 | 0.78 | 13 | 13 | 0.44 | 0.46 | –0.16 | –0.22 | 1.00 | 0.80 |
| T32S_Q_60m | 0.79 | 0.73 | 10 | 10 | 0.33 | 0.28 | –0.13 | –0.04 | 0.99 | 1.02 |
| T32S_Q_max | 0.82 | 0.80 | 11 | 12 | 0.34 | 0.33 | –0.13 | –0.07 | 0.96 | 1.03 |
| T32S_QV_ppP | 0.70 | 0.67 | 11 | 12 | 0.46 | 0.46 | –0.26 | –0.18 | 0.87 | 0.79 |

[a] mean NSE was calculated after setting NSE of individual events to –1 if NSE was lower than –1, to avoid large influence from negative NSE values.

**Changes in manuscript:** changes according to the previous two paragraphs.

**Recommendation**

**(25) Referee's comment:** In its current form the manuscript is not in my mind publishable in HESS. The following major changes would be required:

**Authors' response:** we believe that the major changes requested by the referee can be implemented in a new version of the manuscript, as detailed for the individual comments.

**Changes in manuscript:** see individual points below.

**(26) Referee's comment:** A more informative description of the hydrometeorological data to allow the readers to understand differences between different calibrations

**Authors' response:** see our response to comment #13 above.

**Changes in manuscript:** Include table C1 (supplement) in the manuscript's methods section.

**(27) Referee's comment:** A better justified reasoning for inclusion/exclusion of different error sources

**Authors' response:** this is addressed in our response to points 17-21, 24, 28.

**Changes in manuscript:**

**(28) Referee's comment:** Most importantly, a clear statement about the scientific novelty value of the manuscript where it becomes obvious what are the new findings over just showing that different calibration data lead to different model parameter values and validation performance

**Authors' response:** aspects to highlight in the conclusion and abstract:

- Two-stage calibration is faster, and can provide some performance benefits: e.g. better match of flow volume and peak flow in validation phase.
- Benefits of two-stage calibration are stronger for the LR model.
- Confirmation of earlier findings regarding input and calibration data from Dotto et al. (2014) and Kleidorfer et al. (2009) for a different data set and site (more green area). Findings are independent of the calibration event selection which provides support for their general applicability.

**Changes in manuscript:** changes according to the response above.

**Technical corrections**

**(29) Referee's comment:** Mostly technical comments. The comment for Figure 4 also relates to the content of the manuscript.

**(30) Referee's comment:** Figure 1. Remove the text below the figure (1 map catchment.png). Increase the font size/figure resolution. The legend is hard to read.

**Authors' response:** The text below the figure is added automatically by the Copernicus Latex template used for the submission and would not appear in the final published version of the article. This applies to the other figures as well.

The font size in the legend can be increased.

**Changes in manuscript:** increase font size in legend.

**(31) Referee's comment:** Remove the text below the figure (2 example hydrographs run130.pdf).

**Authors' response:** see above.

**Changes in manuscript:** -

**(32) Referee's comment:** Figure 3. Remove the text below the figure (3 VE PFR histograms.pdf). In the figure caption it is stated peak flow ratios to be on the left whereas in the figure the left panel shows the volume error. Please correct.

**Authors' response:** -.

**Changes in manuscript:** upon further consideration the section and figure in question do not contribute much to the papers goals and they have therefore been removed.

**(33) Referee's comment:** Figure 4. Remove the text below the figure. It is hard to interpret with the given information what is causing the negative NSE for the right panel. Is there a timing difference invisible to the eye? Why does the modelled flow stay at zero for the beginning of the event? Clearly there is rain (left panel), so is the diminished rainfall multiplier and/or increased depression storage value causing all rain falling on the directly connected impervious area to be captured in the depression storage?

**Authors' response:** The main reason why NSE is low is that the low flow rates in the event mean that the variance of observations is low, see also section 2.5, lines 13-14. For the baseline run (left panel), the variance of the observation is 1.2 $L^2$ $s^{-2}$ while for the right panel it is just 0.33 $L^2$ $s^{-2}$. The variance of the errors meanwhile is 0.25 $L^2$ $s^{-2}$ (left) resp. 0.38 $L^2$ $s^{-2}$ (right). NSE is calculated as NSE = 1 – (var err. / var obs.) so the variance of the observations is used as a scaling factor and it is mainly the difference in this factor that causes the degradation in NSE in this example.

**Changes in manuscript:** Add the variance of observations in figure and refer to section 2.5 in the caption for explanation of NSE and discuss this in the text of section 3.1.4, page 12, line 7.

**(34) Referee's comment:** Figure 5, 6, 7, 8. Remove the text below the figure.

**Authors' response:** see above.

**Changes in manuscript:** -

**(35) Referee's comment:** Table 11. Mistake in the NSE single-stage value for D_prec (0.41)? The corresponding value in Table 6 is 0.43?

**Authors' response:** the correct value is 0.43.

**Changes in manuscript:** correct this to 0.43.

Barco, J., Wong, K.M., Stenstrom, M.K., 2008. Automatic Calibration of the U.S. EPA SWMM Model for a Large Urban Catchment. Journal of Hydraulic Engineering 134, 466–474. https://doi.org/10.1061/(ASCE)0733-9429(2008)134:4(466)

Broekhuizen, I., 2019. Uncertainties in rainfall-runoff modelling of green urban drainage systems: Measurements, data selection and model structure (Licentiate thesis). Luleå University of Technology, Luleå.

Deletic, A., Dotto, C.B.S., McCarthy, D.T., Kleidorfer, M., Freni, G., Mannina, G., Uhl, M., Henrichs, M., Fletcher, T.D., Rauch, W., Bertrand-Krajewski, J.L., Tait, S., 2012. Assessing uncertainties in urban drainage models. Physics and Chemistry of the Earth, Parts A/B/C 42–44, 3–10. https://doi.org/10.1016/j.pce.2011.04.007

Dotto, C.B.S., Kleidorfer, M., Deletic, A., Rauch, W., McCarthy, D.T., 2014. Impacts of measured data uncertainty on urban stormwater models. Journal of Hydrology 508, 28–42. https://doi.org/10.1016/j.jhydrol.2013.10.025

Elliott, A., Trowsdale, S., 2007. A review of models for low impact urban stormwater drainage. Environmental Modelling & Software 22, 394–405. https://doi.org/10.1016/j.envsoft.2005.12.005

Fletcher, T.D., Andrieu, H., Hamel, P., 2013. Understanding, management and modelling of urban hydrology and its consequences for receiving waters: A state of the art. Advances in Water Resources 51, 261–279. https://doi.org/10.1016/j.advwatres.2012.09.001

Houska, T., Kraft, P., Chamorro-Chavez, A., Breuer, L., 2015. SPOTting Model Parameters Using a Ready-Made Python Package. PLOS ONE 10, e0145180. https://doi.org/10.1371/journal.pone.0145180

Kleidorfer, M., Deletic, A., Fletcher, T.D., Rauch, W., 2009. Impact of input data uncertainties on urban stormwater model parameters. Water Science and Technology 60, 1545–1554. https://doi.org/10.2166/wst.2009.493

Krebs, G., Kokkonen, T., Valtanen, M., Setälä, H., Koivusalo, H., 2014. Spatial resolution considerations for urban hydrological modelling. Journal of Hydrology 512, 482–497. https://doi.org/10.1016/j.jhydrol.2014.03.013

Petrucci, G., Bonhomme, C., 2014. The dilemma of spatial representation for urban hydrology semi-distributed modelling: Trade-offs among complexity, calibration and geographical data. Journal of Hydrology 517, 997–1007. https://doi.org/10.1016/j.jhydrol.2014.06.019

Rossman, L.A., Huber, W.C., 2016. Storm Water Management Model Reference Manual. Volume I: hydrology (Revised). U.S. Environmental Protection Agency, Cincinnati.

Sun, N., Hall, M., Hong, B., Zhang, L., 2014. Impact of SWMM Catchment Discretization: Case Study in Syracuse, New York. Journal of Hydrologic Engineering 19, 223–234. https://doi.org/10.1061/(ASCE)HE.1943-5584.0000777

Tscheikner-Gratl, F., Zeisl, P., Kinzel, C., Leimgruber, J., Ertl, T., Rauch, W., Kleidorfer, M., 2016. Lost in calibration: why people still do not calibrate their models, and why they still should – a case study from urban drainage modelling. Water Science and Technology 74, 2337–2348. https://doi.org/10.2166/wst.2016.395

Note to reviewers: most of the tables in the manuscript have been updated since the previous version of this manuscript. The automated tool used to create this version with changes marked struggles to display this in an intelligible way, so for most tables just the new version is shown. This is indicated in the table captions as well.

[revised manuscript text omitted]

---

## Author Response (AR2)

Reviewer's comment: The authors have provided a thorough response to the first round review comments, and at parts revised the manuscript accordingly. Leaving out some of the content present in the original manuscript has improved the clarity of the revised manuscript. I my view there would be still be room for a more concise presentation to highlight the main findings of the research.

Documentation of the two-stage calibration (impermeable vs. permeable parameters) deserves to be published to a wider audience, currently the results are in my understanding only available in the licentiate thesis of the first author and in a brief form in a conference extended abstract.

We would like to thank the reviewer for once again taking the time to review the paper and provide feedback. We accept the reviewer's point about the conciseness of the article and have responded by making the following adjustments:

- The title has been changed to explicitly mention the two-stage approach to calibration: "Event selection and two-stage approach for calibration of green urban drainage models"
- The perturbation of flow data in the calibration phase has been removed entirely, except for a short mention when discussing the rainfall multipliers (p 13 l 7-8).
- The question of model resolution per se has been made less prominent, especially in the introduction and the conclusions. However, we still think it is interesting to show some results from this exercise, since the two-stage calibration shows clear benefits in the validation phase of the low-resolution model.
- The subsection on "other sources of uncertainty" has been removed from the materials and methods section, since it is now mostly superfluous. The description of the low-resolution model has been included in Sect. 2.2 'Runoff model and calibration approach'.

Reviewer's comment: The authors have retained the analysis on +/- 40% perturbed discharge data and the level of model discretization. In my view, as already expressed in my first review, the results related to these analyses do not have much novelty value and are partly trivial. The authors argue the novelty by claiming that earlier studies have predominantly addressed impervious catchments instead of 'greener' urban catchments. I am not totally convinced this is the case. In page 2 lines 20-21 the authors refer to three earlier studies with regard to spatial model resolution assessments, and state below that 'these investigations used predominantly impervious catchments'. Strictly speaking, this is true, but I find it a bit misleading as out of six catchments addressed in the referenced studies two represent the same level of imperviousness as the study area in the current manuscript. Catchment 3 of Krebs et al., 2014, imperviousness 19 % (p. 483, Section 2 Study site and data) and Sucy catchment of Petrucci and Bonhomme, 2014, imperviousness ~20% (p. 1005, Figure 6b).

(Please note that the lines in question have been removed from the text in the revised version.) The reviewer is right in their clarification of the characteristics of the catchments used in previous studies.

Reviewer's comment: Also, the authors point out that the variability of the calibrated rainfall multiplier values is larger for low-resolution than for high resolution models (p. 17, lines 1-2) and attribute this to 'rainfall multipliers appearing to behave in a more physical way' (p. 17, line 9). This sounds like an obvious result as the low resolution models, unlike the high resolution models, have another parameter directly affecting the volume of runoff ('Percentage runoff routed from impervious to pervious', Table 3, p. 10). In page 14 lines 6-8 the authors state themselves that while

in Dotto et al. (2014) the mismatch between rainfall and discharge volumes was controlled by the calibrated value of percentage imperviousness in the current paper the same impact is obtained by manipulating rainfall multiplier values. Now in low-resolution models of the current manuscript, in essence, both of these controls are present (rainfall multipliers and 'Percentage runoff routed from impervious to pervious'), which leads to a larger variability in any of these two controls individually, which is not surprising.

We accept the reviewer's point about this issue. In response to other comments, some of the results relating to the LR model have been removed from the manuscript, including this part.

Reviewer's comment: I would like to see this manuscript published but I feel a bit frustrated to see the most valuable part of the research results being mixed with the rather inconclusive results related to other sources of uncertainty. The latter are often trivial or confirm earlier findings of other research. In my mind, the manuscript would certainly benefit from a tighter framing and focus. In its current state most of the Conclusions address the results related to performance differences between one-stage and two-stage calibrations, or alternatively the relatively obvious results about the relationships between rainfall multipliers and the scaling coefficient of flow series (or variation of rainfall multipliers between high- and low-resolution models). Hardly any conclusions are drawn on the selection of calibration events (P_sum, PI_mean, PI_30m etc.), which according to the title of the manuscript should be in the focus. In my view this should be fixed by changing the title to emphasize the one-stage / two-stage calibrations and revising the contents accordingly. Should the authors wish to keep the original title, then the main results and conclusions should be aligned with it.

As explained above we have removed the parts of the manuscript focusing on checking/confirming earlier findings and focused more on the different calibration strategies and the two-stage approaches. The two-stage approach is now mentioned in the new title of the paper and covered more extensively in the Introduction. The conclusions have been revised to focus on various calibration strategies (although it is difficult to say which CS is the best, some CSs can be said to be worse than the others) and the effects of the two-stage calibrations.

Specific (technical) comments

Reviewer's comment: p. 6, line 4: Correct the reference Fuentes-Andino…

This has been corrected.

Reviewer's comment: p.7, line 18: "..values of certain event characteristic…", vague language, replace with e.g. 'given event characteristic'

This phrasing has been improved: "The single-stage CSs used the six events with the highest values for a given event characteristic…"

Reviewer's comment: p. 8, line 4: Not totally clear what 'Doing this' refers to. Consider rephrasing the sentence.

Rephrased to (now p8, line 25-26): "Adjusting which calibration stage the events are available for in the calibration procedure (in a manner that is consistent for all events) would require …"

Reviewer's comment: p. 11, line 23: Word 'Figure' missing from Figure 2.

This has been fixed.

Reviewer's comment: p. 17, line 18: "When switching from the high resolution to the low-resolution model the single-stage CSs were no longer able to predict up to 5 events" Consider rephrasing, hard to follow a sentence.

This sentence has been rephrased (p15 line 30-31): "For the single-stage CSs the low-resolution model predicted up to five fewer events satisfactorily than the high resolution model"

Reviewer's comment: p. 19, lines 10-11: "inter-CS variation for the same events varies from 0.15 to 1.25." The difference of 1.25 is hard to see from Figure 6, as the y-scale is cut at zero.

Although we accept the reviewers point, we still prefer to cut the y-axis in the figure. The reason for this is that it would make differences among the positive NS values more difficult to see, and it is (in our view) not so interesting to show the variation among different CSs, when they all have bad performance.

Reviewer's comment: p. 21, line 7: "more less consistent than.." should probably read 'less consistent result'

In response to other comments by the reviewer, this section of the text has been removed entirely.

Reviewer's comment: p. 22, Table 8: not obvious what all column titles mean, what is 'Clip mean NSE'?

Footnotes have been added to table 6 and 7 (numbers in new version of manuscript) to explain the column headers.

Finally, we realized while making the revisions that "calibration strategy" and "calibration scenario" were used interchangeably. We have now changed this to "calibration strategy" throughout.

[revised manuscript text omitted]

---

## Author Response (AR3)

Dear Authors,

I am generally happy with the revised manuscript, in my opinion it offers now a much crisper presentation of the most valuable results of this research. I recommend acceptance of the manuscript with some minor changes/clarifications.

Title: Consider rephrasing the title, 'calibration of green urban drainage models' sounds a bit strange. What is a 'green model'?

The title has been rephrased: "Event selection and two-stage approach to calibrating models of green urban drainage systems"

P. 3, lines 23-24: "This two-stage calibration has not been investigated for urban drainage models,…" Refer to the licentiate thesis / conference abstract where this idea was introduced.

A reference to the conference abstract has been added, since this is where the idea was introduced, and the licentiate thesis merely summarizes it.

P. 3, line 27: "…of the paper that follows is…" Perhaps just 'this paper' instead of 'the paper that follows'?

This has been changed.

Table 1 (and elsewhere): Should QV rather be runoff depth as its unit is mm?

QV is indeed expressed in mm, but we prefer to refer to it as "runoff volume", mainly because discussing "volume error" is somewhat more intuitive than referring to "runoff depth error". A clarification has been added in table 1 that runoff volume is expressed as mm of runoff per catchment area.

P. 8, lines 9-12: Remove parentheses. Entire sentences should not appear enclosed in parentheses in the middle of the text. Rephrase paragraph if needed.

The sentence (without parentheses) has been moved after the sentence "Therefore, these events were suitable for calibration of impervious area parameters in the first stage of the calibration process." so that it fits better in the paragraph.

P. 10, lines 24-26: Do the cited articles present results on one-stage vs. two-stage calibrations? I presume not. Rephrase the text so that it does not sound like the cited research would confirm results on one-stage/two-stage calibrations.

This sentence has been rephrased: "Considering the errors in total runoff volume, the two-stage CSs performed better for the HR model. However, for the LR model (where runoff volumes were higher in general, as also reported by Tscheikner-Gratl (2016)), the single stage calibrations had smaller volume errors. The changes in volume errors between the HR and LR model were similar to earlier findings by Krebs et al. (2016)."

P. 11, last line: Remove parentheses. Rephrase if needed.

This has been done and slightly rephrased: "This runoff was caused by impervious areas draining to green areas. The runoff from green areas was <5% of the total simulated runoff volume for 4 model runs, <10% for an additional 3 runs, and 11.6%, 11.7%, 21.7%, 22.9% and 25.7% respectively for 5 additional runs. Note that with 6 CSs with 3 first-stage events each, there were 18 model runs in total. The last mentioned 5 runs concerned 3 different events with a percentage runoff (calculated before applying rainfall multipliers) between 11% and 12%."

P. 13, line 2: "..from 0.48 to 2.92,…" Consider adding a couple of sentences to further elaborate this. I do not think the options you offer are necessarily equally likely. This probably is not a result of a (systematic) flow gauging error as it varies from an event to another. It well could be, as you suggest, attributed to differences between the gauge rainfall and the catchment mean rainfall. Alternatively it could result e.g. from the model's deficiency in describing correctly losses/stored water for different events.

The discussion of this has been extended: "The values of rainfall multipliers found in the calibration process ranged from 0.48 to 2.92, indicating a mismatch between the observed rainfall and the rainfall that allows for the best fit of the simulated runoff to observed runoff. Several factors may contribute to this. First, underestimation of rainfall or underestimation of runoff by the respective sensors may lead to higher rainfall multipliers and vice versa. Errors in the size of (sub)catchments may also influence this. Second, the gauge rainfall may not match the catchment-averaged rainfall due to the spatial variability of the rainfall. Thirdly, some deficiencies in the model may be compensated for, to some extent, by adjusting the rainfall multiplier. Without further investigations it is not possible to distinguish between different factors influencing the values of rainfall multipliers."

P. 13, lines 7-8: "Secondly, decreasing or increasing all flow rates by 40% prior to calibration changed the average rainfall multipliers by -37% and +33% respectively." I still think this is rather obvious and should be omitted.

Although we see the reviewers point that this is not an unexpected result, it does help show that the rainfall multipliers work as expected, which is why we would prefer to still mention it in the article.

P. 15, lines 3-4: "…events, but produced wrong timing, which was also reflected in the peak flow and volume errors." Not sure I understand this. Form the way PFR was defined (p. 9, lines 21-23) it seems that timing should not be an issue as the event peak flow value is picked at the time it occurred and hence a timing difference between modelled and observed flow should not be important? The same applies for flow volume – timing is not an issue as long as the time window encloses the entire event for both modelled and observed values?

What was meant was that the values for peak flow ratio and flow volume were good, which further supports the good visual agreement between the hydrographs. The sentence has been rephrased to clarify: "For the other CSs, examination of the hydrographs showed that they predicted well the peak flow and the total runoff volume of the events, but produced wrong timing compared to the observed hydrograph."

P. 15, lines 20-21: "…for most events flows were underestimated, which may be (at least partially) attributed to the discrepancies between observed rainfall and flow found in the calibration phase (see Sect. 3.2.2)." Not sure I understand this either. The idea of introducing rainfall multipliers was to fix the potential mismatch between gauge rainfall and catchment mean rainfall. Now why would the discrepancy between gauge rainfall and catchment rainfall result in consistently underestimating (and not overestimating) the flow? Please clarify.

The rainfall multipliers (when their value is >1) increase the rainfall input into the model, thereby increasing the runoff volumes. The average multiplier in the calibration phase was 1.2. No multiplier was applied in the validation phase, which means that the rainfall volume was not increased and the corresponding increase in runoff was not present, which can explain why underestimation of flows was the more common problem in the validation phase. The sentence has been rephrased to clarify this: "The results for both total volumes and peak flows indicate that for most events flows were underestimated, which may be (at least partially) attributed to the need to multiply the observed rainfall by (on average) a factor of 1.2. to best match the observed flows during the calibration phase (see Sect. 3.2.2); such adjustment was not applied in the validation phase"

P. 15, line 24: "inter-CS variation for the same events varies from 0.15 to 1.25." What are these values? Not quite clear to me.

This refers to the difference in NSE between the best and worst CS for the same event. The sentence has been rephrased to clarify: "When examining the NSE of the validation events (see the bottom panel of Error! Reference source not found.), more variation among the different CSs became visible,

although the amount of variation was still event-dependent: the difference (in NSE) between the best and worst CS for the same event varied from 0.15 to 1.25."

P. 15, lines 31-33: "…model, while from the two-stage CSs only T32S_D_prec satisfactorily predicted fewer events, and T32S_P_sum, T32S_Q_max, and T32S_QV_ppP actually predicted a higher number of events satisfactorily." This probably refers to comparison between HR and LR models, but in my mind it is not quite clear. Consider rephrasing.

This has been rephrased: "For the single-stage CSs the low-resolution model predicted up to five fewer events satisfactorily than the high-resolution model, while from the two-stage CSs only T32S_D_prec satisfactorily predicted fewer events with the LR model than with the HR model, and T32S_P_sum, T32S_Q_max, and T32S_QV_ppP actually predicted more events satisfactorily with the HR model."

P. 16, line 3: "…with this metric: First,…" replace the colon with a full stop.

This has been changed.

P. 18, line 4: "…, the NSE…" Should this read 'the event mean NSE', i.e. does it refer to the mean NSE value over all calibration events (instead of NSE values over individual events)? If you change the wording, check also elsewhere (e.g. 3.5, Conclusions) for consistency.

This does indeed refer to the mean NSE over all calibration events. The wording has been changed throughout the paper to read "event mean NSE" where applicable.

P. 19, lines 7-8. " By contrast, in the validation phase the NSE was better for the single-stage CSs." This refers to the event mean NSE? But not to the 'Joint NSE' (Table 6)? Should this be somehow discussed? At times it is difficult for the reader to follow which performance criterion is referred to, in particular as they can give mixed messages.

This refers to the event mean NSE and this has been clarified in the text. The joint NSE is not discussed here because it may mask the fact that some events are poorly predicted. E.g. in Table 6 it can be seen that all CSs have a joint NSE of >0.5, even though they all have a negative NSE for at least two events. It is particularly noticeable for T32S_P_sum, which has the (shared) 3rd highest joint NSE, despite having 5 events with negative NSE.  This drawback of the joint NSE has been added to section 3.3.2, p17, l 6-7.

P. 20, line 4: "NSE values" Does this mean 'event mean NSE values'?

Yes, and this has been changed to say "event mean NSE" in the text.

[revised manuscript text omitted]